# Distinct mitochondrial defects trigger the integrated stress response depending on the metabolic state of the cell

Eran Mick[1,2,3†‡], Denis V Titov[1,2,3§], Owen S Skinner[1,2,3], Rohit Sharma[1,2,3], Alexis A Jourdain[1,2,3], Vamsi K Mootha[1,2,3]*

[1]Howard Hughes Medical Institute and Department of Molecular Biology, Massachusetts General Hospital, Boston, United States; [2]Broad Institute, Cambridge, United States; [3]Department of Systems Biology, Harvard Medical School, Boston, United States

*For correspondence:
vamsi@hms.harvard.edu

Present address: †Division of Infectious Diseases & Division of Pulmonary and Critical Care Medicine, Department of Medicine, University of California, San Francisco, San Francisco, United States; ‡Chan Zuckerberg Biohub, San Francisco, United States; §Department of Nutritional Science and Toxicology, Department of Molecular and Cell Biology & Center for Computational Biology, University of California, Berkeley, United States

**Abstract** Mitochondrial dysfunction is associated with activation of the integrated stress response (ISR) but the underlying triggers remain unclear. We systematically combined acute mitochondrial inhibitors with genetic tools for compartment-specific NADH oxidation to trace mechanisms linking different forms of mitochondrial dysfunction to the ISR in proliferating mouse myoblasts and in differentiated myotubes. In myoblasts, we find that impaired NADH oxidation upon electron transport chain (ETC) inhibition depletes asparagine, activating the ISR via the eIF2$\alpha$ kinase GCN2. In myotubes, however, impaired NADH oxidation following ETC inhibition neither depletes asparagine nor activates the ISR, reflecting an altered metabolic state. ATP synthase inhibition in myotubes triggers the ISR via a distinct mechanism related to mitochondrial inner-membrane hyperpolarization. Our work dispels the notion of a universal path linking mitochondrial dysfunction to the ISR, instead revealing multiple paths that depend both on the nature of the mitochondrial defect and on the metabolic state of the cell.

## Introduction

The mitochondrial electron transport chain (ETC) supports myriad, mechanistically linked functions. It provides the driving force for ATP synthesis by oxidative phosphorylation (OXPHOS) (*Mitchell, 1961*; *Mitchell, 2011*), maintains redox balance of cofactor pairs (NADH:NAD$^+$, CoQH$_2$:CoQ) that are coupled to hundreds of cellular reactions (*Williamson et al., 1967*; *Ying, 2008*; *Santidrian et al., 2013*; *Titov et al., 2016*; *Xiao et al., 2018*; *Goodman et al., 2018*; *Mitchell, 1975*; *Turunen et al., 2004*; *Alcázar-Fabra et al., 2016*), contributes to the electrochemical gradient that drives transport of proteins and metabolites across the mitochondrial inner-membrane (*LaNoue and Schoolwerth, 1979*; *Martin et al., 1991*), and is a major determinant of cellular oxygen levels and tolerance (*Taivassalo et al., 2002*; *Campian et al., 2004*; *O'Hagan et al., 2009*; *Brand, 2016*; *Jain et al., 2019*). Defects in the ETC and OXPHOS system are associated with a spectrum of human pathology ranging from individually rare genetic disorders to common conditions, such as neurodegeneration (*Schapira, 2008*; *Hu and Wang, 2016*; *Lin and Beal, 2006*; *Area-Gomez et al., 2019*), diabetes (*Kelley et al., 2002*; *Mootha et al., 2004*; *Ritov et al., 2005*; *Ritov et al., 2010*), select forms of cancer (*Wallace, 2012*; *Reznik et al., 2017*; *Gaude and Frezza, 2014*) and the aging process itself (*Sun et al., 2016*; *Bratic and Larsson, 2013*).

Mutations in nearly 300 nuclear or mitochondrial (mtDNA) genes have been implicated in mitochondrial disorders that affect at least 1 in 4300 live births and lack effective treatments (*Schapira, 2012*; *Vafai and Mootha, 2012*; *Gorman et al., 2015*; *Gorman et al., 2016*; *Frazier et al., 2019*). Management of these conditions is complicated by their striking heterogeneity. They can

develop acutely or progressively and can impact multiple organ systems or just a single cell type. Disease often manifests in post-mitotic tissues, such as skeletal muscle or the nervous system, but this pattern does not simply correlate with ATP demand or tissue expression of the causal gene. Deciphering how cells sense and respond to mitochondrial dysfunction is thus a major challenge with the potential to inform the common conditions that exhibit a decline in mitochondrial health.

Studies in patients, animals and cultured cells have consistently identified a gene expression program associated with the integrated stress response (ISR) as a signature of mitochondrial dysfunction (*Martinus et al., 1996*; *Zhao et al., 2002*; *Kühl et al., 2017*; *Celardo et al., 2017*; *Quirós et al., 2017*; *Fujita et al., 2007*; *Silva et al., 2009*; *Tyynismaa et al., 2010*; *Dogan et al., 2014*; *Bao et al., 2016*; *Magarin et al., 2016*; *Lehtonen et al., 2016*; *Khan et al., 2017*). The ISR is triggered by various insults, including nutrient deficiency, unfolded protein stress and pathogen infection. In mammalian cells, it is typically activated by a family of four kinases, each of which can phosphorylate the alpha subunit of translation initiation factor 2 (eIF2$\alpha$) in response to distinct stimuli. eIF2$\alpha$ phosphorylation acutely inhibits global protein synthesis but promotes translation of transcription factors, such as activating transcription factor 4 (ATF4) and DNA damage-inducible transcript three protein (DDIT3), that then engage their downstream targets (*Harding et al., 2003*; *Palam et al., 2011*; *Harding et al., 2000*; *Vattem and Wek, 2004*; *Wek, 2018*; *Pakos-Zebrucka et al., 2016*; *Taniuchi et al., 2016*).

The transcriptional program induced by the ISR varies across species and cell types and depends on the underlying trigger. Nevertheless, it frequently encompasses amino acid transport and biosynthesis genes, cytosolic tRNA synthetases and translation factors, pro-apoptotic factors, and genes involved in antioxidant defense, proteostasis and organelle quality control (*Harding et al., 2003*; *Han et al., 2013*). In tissues affected by mitochondrial disease, the ISR activates the mitochondrial 1-carbon pathway (*Kühl et al., 2017*; *Tyynismaa et al., 2010*; *Bao et al., 2016*; *Nikkanen et al., 2016*) and is thought to underlie secretion of the circulating cytokines fibroblast growth factor 21 (FGF21) and growth/differentiation factor 15 (GDF15) that are under consideration as disease biomarkers (*Lehtonen et al., 2016*; *Fujita et al., 2015*; *Yatsuga et al., 2015*; *Chung et al., 2017*; *Miyake et al., 2016*; *Restelli et al., 2018*). It remains unclear whether the ISR plays a protective role in mitochondrial disease or rather contributes to pathology (*Khan et al., 2017*; *Suomalainen and Battersby, 2018*; *Lamech and Haynes, 2015*).

How mitochondrial dysfunction is sensed to trigger the ISR is a major open question. Defects in mtDNA maintenance and expression are most frequently associated with ISR activation in vivo (*Kühl et al., 2017*; *Lehtonen et al., 2016*). However, such defects are expected to impose multiple biochemical consequences on the ETC/OXPHOS system, which are difficult to disentangle. To address this challenge, we profiled global gene expression, bioenergetics and metabolism in muscle cells that were proliferating (myoblasts) or differentiated (myotubes) and acutely treated with a panel of small-molecule mitochondrial inhibitors. To isolate the specific biochemical pathways that trigger the ISR, we employed a suite of 'protein prostheses' that facilitate NADH recycling to NAD$^+$, in the cytosol or in mitochondria, independent of the ETC (*Titov et al., 2016*).

Our systematic analysis reveals that multiple mechanisms can link mitochondrial dysfunction to ISR activation. In proliferating myoblasts, we show that ETC inhibition elevates the mitochondrial and cytosolic NADH/NAD$^+$ ratios, hindering aspartate synthesis and ultimately depleting asparagine, which activates the ISR via the eIF2$\alpha$ kinase GCN2. In myotubes, we find that inhibition of ATP synthase activates the ISR via a distinct mechanism related to mitochondrial inner-membrane hyperpolarization. Our results reject the notion of a universal trigger of the ISR in mitochondrial dysfunction, and rather, reveal multiple paths to its activation that depend both on the nature of the mitochondrial defect and on the metabolic state of the cell.

## Results

We sought to interrogate the genomic response to mitochondrial dysfunction and its dependence on the metabolic state of the cell. To this end, we utilized proliferating C2C12 mouse myoblasts, which can be induced to exit the cell cycle and then terminally differentiate into myotubes (*Yaffe and Saxel, 1977*; *Blau et al., 1985*; *Figure 1A*). Differentiation triggers mitochondrial biogenesis and promotes oxidative metabolism as the cells shift from a primary focus on nutrient uptake and biosynthesis to self-maintenance and specialized functions (*Moyes et al., 1997*; *Leary et al.,*

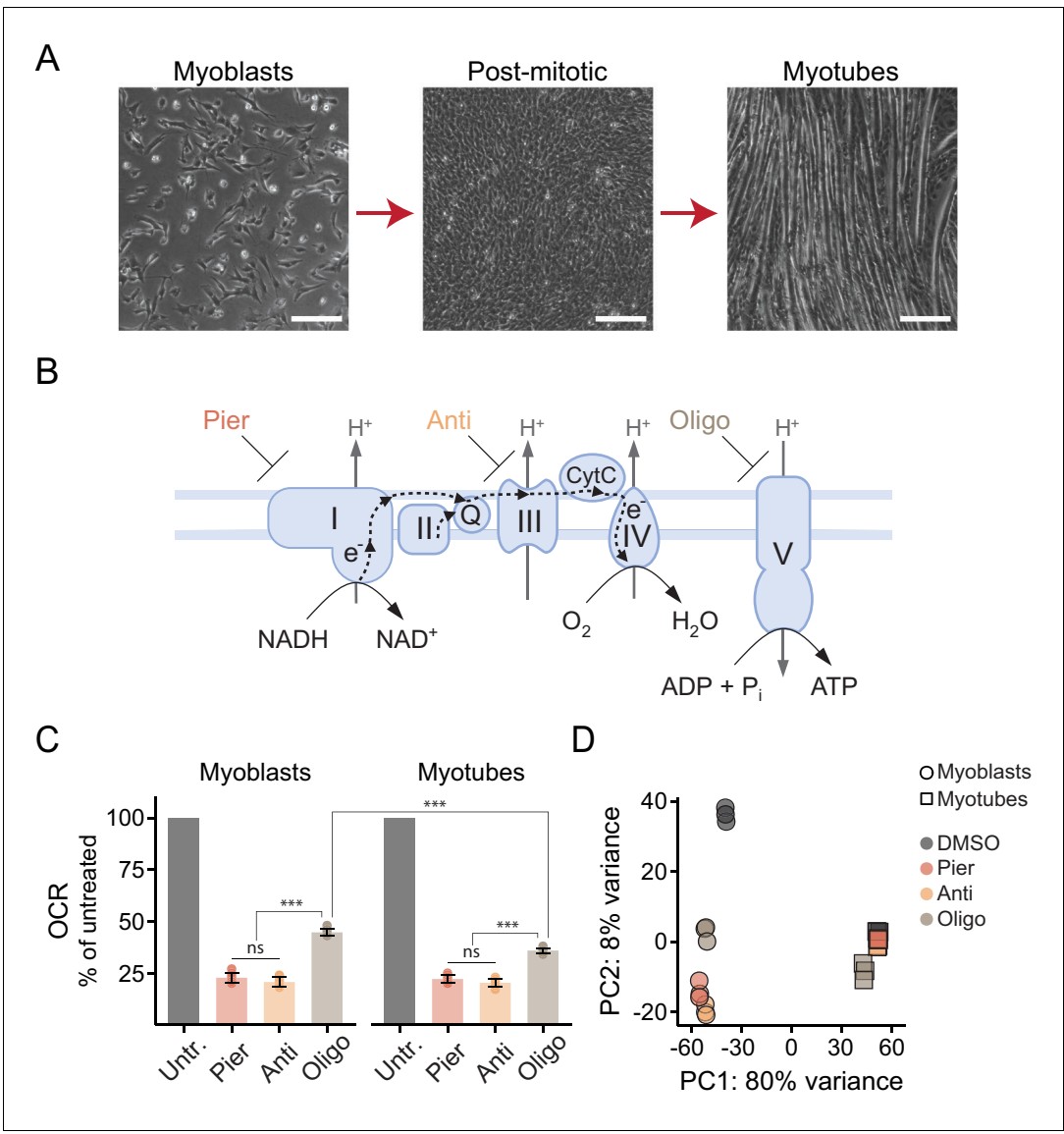

**Figure 1.** Interrogating the genomic response to mitochondrial dysfunction in proliferating and differentiated muscle cells. (**A**) Phase contrast images of proliferating C2C12 myoblasts (left), post-mitotic cells cultured in low-serum media for 24 hr following confluence (middle), and differentiated myotubes (right). Scale bar denotes 200 µm. (**B**) Schematic of the ETC/OXPHOS system and the site of action of small-molecule inhibitors. Pier, piericidin; Anti, antimycin; Oligo, oligomycin. (**C**) Oxygen consumption rate (OCR) of myoblasts and myotubes acutely treated with mitochondrial inhibitors, normalized in each well to OCR prior to treatment (Untr., untreated). Mean ± SD, N = 7–9 from two experiments. The Games-Howell test was used for all pairwise comparisons. ns, p>0.05; ***, p<0.001. (**D**) Principal component analysis (PCA) of gene expression levels derived from RNA-seq in myoblasts and myotubes treated with inhibitors for 10 hr. N = 3.

*1998*; *Casadei et al., 2009*; *Remels et al., 2010*; *Sin et al., 2016*; *Vander Heiden et al., 2009*; *Lemons et al., 2010*).

We acutely treated myoblasts and myotubes with inhibitors of complex I (piericidin), complex III (antimycin) or ATP synthase (oligomycin) (*Figure 1B*) in media lacking pyruvate and uridine, to avoid masking known metabolic vulnerabilities of ETC-compromised cells (*Morais et al., 1980*; *King and Attardi, 1989*) (Materials and methods). We confirmed the inhibitors exerted the expected effects on respiration (*Figure 1C*). Complex I or complex III inhibition abrogated mitochondrial oxygen consumption while cells treated with the ATP synthase inhibitor retained residual respiration, not coupled to ATP synthesis, often termed oligomycin-resistant respiration or leak respiration

(*Jastroch et al., 2010*; *Divakaruni and Brand, 2011*). Myotube mitochondria showed less leak respiration, indicative of tighter coupling (*Sin et al., 2016*).

We next performed RNA-sequencing to profile global gene expression changes following 10 hr of treatment, roughly corresponding to the duration of the myoblast cell cycle. Principal component analysis (PCA) of the combined data from myoblasts and myotubes suggested marked differences in their response to mitochondrial inhibition but the strongest trend was the fundamental distinction between the cell states (*Figure 1D*). We therefore proceeded to analyze the results separately for myoblasts and for myotubes.

## Mechanistically distinct inhibitors of mitochondria trigger the ISR in myoblasts

PCA of myoblast gene expression revealed a dominant first principal component (PC1; 57% of the variance) along which inhibited cells progressed, with piericidin and antimycin exerting a more pronounced effect than oligomycin (*Figure 2A*; *Supplementary file 1*). To gain insight into genes driving variation along PC1, we subjected the 500 genes with the most positive weights (upregulated along PC1) and, separately, the 500 genes with the most negative weights (downregulated) to *cis*-regulatory analysis (*Janky et al., 2014*) (Materials and methods). We also performed gene set enrichment analysis (GSEA) (*Mootha et al., 2003*; *Subramanian et al., 2005*) of REACTOME pathways (*Croft et al., 2014*) using PC1 weights as the gene ranks (Materials and methods).

Both approaches converged on activation of the ISR as a major trend driving gene expression following inhibitor treatments. *Cis*-regulatory analysis identified binding motifs for the ISR master regulator ATF4 as the most enriched in promoters of genes upregulated along PC1 (*Figure 2B*; *Supplementary file 2*), while GSEA highlighted pathways typically upregulated as part of the ISR, including cytosolic tRNA synthetases and translation factors, amino acid transport and biosynthesis genes, and additional ISR regulators (*Harding et al., 2003*; *Han et al., 2013*; *Figure 2C*).

At the same time, the analyses revealed downregulation of genes linked to cell proliferation. Promoters of downregulated genes were enriched for motifs or ChIP-seq peaks of E2F-family transcription factors, master regulators of the cell cycle (*Johnson et al., 1993*; *Helin, 1998*; *Ren et al., 2002*; *Fischer and Müller, 2017*), and of SREBP-1/2, regulators of cholesterol and lipid metabolism (*Yokoyama et al., 1993*; *Hua et al., 1993*; *Wang et al., 1994*; *Horton et al., 2002*; *Figure 2B*; *Supplementary file 2*). The cell cycle, DNA replication and cholesterol biosynthesis pathways also emerged as the most sharply depressed in the GSEA results (*Figure 2C*). This is consistent with the proliferative defect ETC inhibition is expected to impose in our media conditions.

We generated a volcano plot relating each gene's magnitude and significance of association with PC1 (Materials and methods). We then annotated it with verified target genes of ATF4 and DDIT3 derived from ChIP-seq data (*Han et al., 2013*), and with members of the REACTOME cell cycle and cholesterol pathways (*Figure 2D*; *Supplementary file 1*). The result demonstrated the striking enrichment of these categories among differentially expressed genes. Expression changes of top genes associated with PC1 from the enriched categories further illustrated that the transcriptional signatures of ISR activation (*Figure 2E*) and impaired proliferation (*Figure 2F*) mirrored one another (*Hamanaka et al., 2005*).

A large fraction of ISR target genes displayed the gradation of induction with the different inhibitors reflected along PC1, that is antimycin > piericidin > oligomycin (Group 1 in *Figure 2—figure supplement 1A*). Other targets, however, showed comparable induction across the inhibitors (Group 2) and a minority even showed the reverse gradation (Group 3). These patterns suggest complex cross-regulation of some ISR targets and underscore the non-equivalence of different modes of mitochondrial dysfunction. Nevertheless, the transcriptional signatures we observed were accompanied in all cases by eIF2α phosphorylation and ATF4 protein accumulation (*Figure 2—figure supplement 1B*).

## Oxidizing cytosolic NADH/NAD$^+$ is sufficient to ablate ISR activation by complex I inhibition in myoblasts

In myoblasts, complex I or complex III inhibition triggered the ISR more potently than ATP synthase inhibition. We therefore reasoned that a decrease in ATP levels due to breakdown of OXPHOS was less likely to play the operative role in ISR activation, whereas impaired NADH oxidation ('reductive

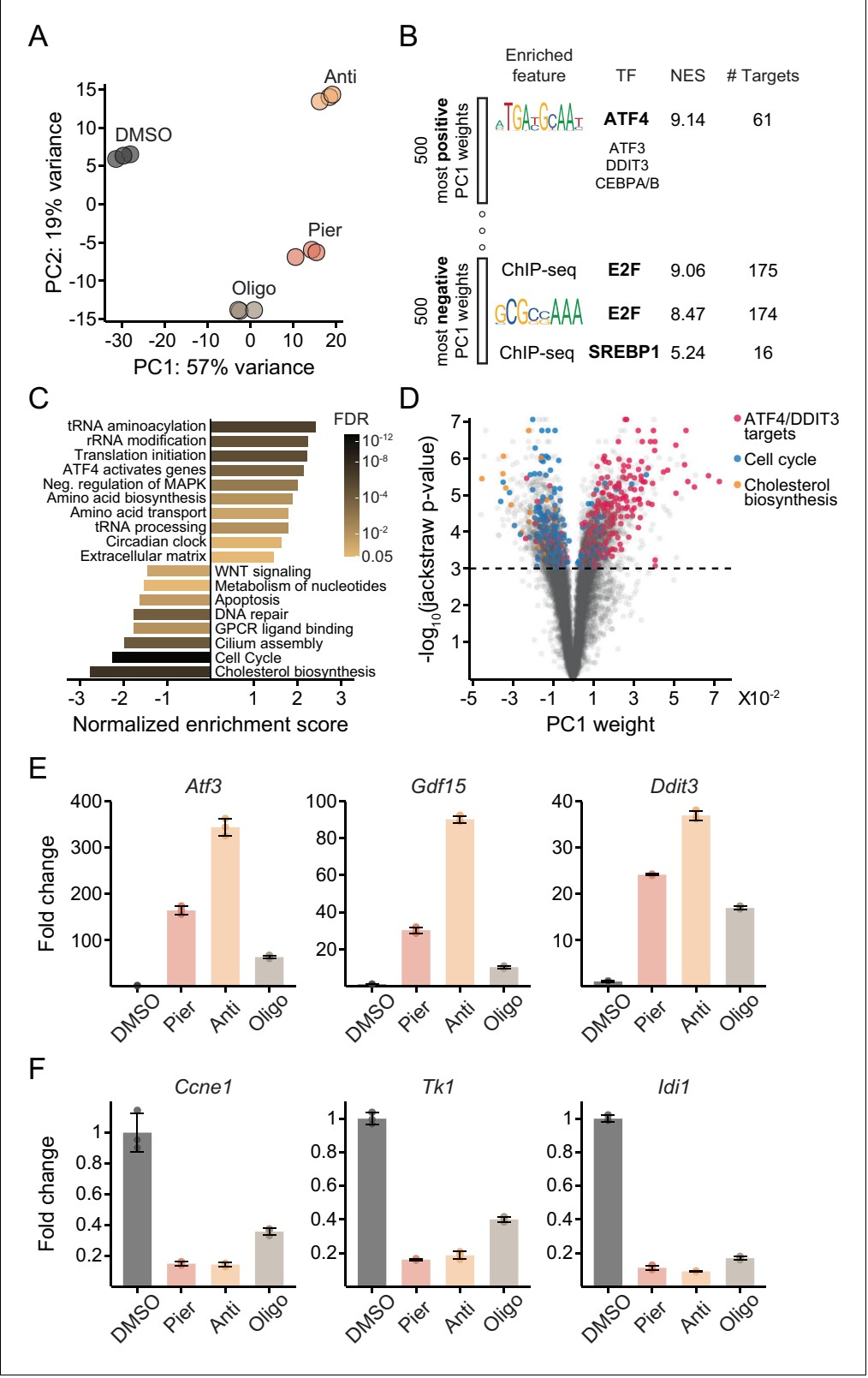

**Figure 2.** Mechanistically distinct mitochondrial inhibitors trigger the ISR and depress proliferative gene expression in myoblasts. (**A**) PCA of gene expression levels derived from RNA-seq in myoblasts treated with inhibitors for 10 hr. N = 3 (same samples as in *Figure 1D*). Pier, piericidin; Anti, antimycin; Oligo, oligomycin. Detailed results are provided in *Supplementary file 1*. (**B**) Enriched features in the promoters of the 500 genes

*Figure 2 continued on next page*

*Figure 2 continued*

with the most positive PC1 weights and the 500 genes with the most negative weights, based on iRegulon analysis of transcription factor binding motifs and ChIP-seq peaks. TF, transcription factor; NES, normalized enrichment score; # Targets, number of gene targets. Detailed results are provided in *Supplementary file 2*. (C) Gene set enrichment analysis of REACTOME pathways using PC1 weights as the gene ranks. Detailed results are provided in *Figure 2—source data 1*. (D) Volcano plot relating each gene's PC1 weight (x-axis) and jackstraw P-value for significance of association with PC1 (y-axis). (E) Fold-change from DMSO derived from RNA-seq for representative ATF4/DDIT3 target genes strongly upregulated along PC1 (*Atf3*, activating transcription factor 3; *Gdf15*, growth/differentiation factor 15; *Ddit3*, DNA damage-inducible transcript 3). Mean ± SD. (F) Fold-change from DMSO derived from RNA-seq for representative cell cycle pathway genes (*Ccne1*, G1/S-specific cyclin-E1; *Tk1*, thymidine kinase, cytosolic) and a cholesterol pathway gene (*Idi1*, isopentenyl-diphosphate delta isomerase 1) strongly downregulated along PC1. Mean ± SD.

The online version of this article includes the following source data and figure supplement(s) for figure 2:

**Source data 1.** Gene set enrichment analysis.

**Figure supplement 1.** Additional data on transcriptional and protein markers of the ISR following mitochondrial inhibitor treatments in myoblasts.

---

stress') presented a more compelling candidate. To explore this possibility, we generated myoblasts expressing *Lb*NOX, a bacterial water-forming NADH oxidase that directly converts NADH to $NAD^+$ independent of the ETC (*Titov et al., 2016*; *Figure 3A*, *Figure 3—figure supplement 1A*). We also generated cells expressing mito*Lb*NOX, a variant of *Lb*NOX targeted to the mitochondrial matrix (*Titov et al., 2016*; *Figure 3A*, *Figure 3—figure supplement 1A*). Finally, we generated cells expressing NDI1, a piericidin-resistant yeast protein that substitutes for complex I in transferring electrons from NADH to coenzyme Q (CoQ) and thus facilitates both continued NADH oxidation and ATP synthesis by OXPHOS (*Titov et al., 2016*; *De Vries et al., 1992*; *Seo et al., 1998*; *Seo et al., 2000*; *Figure 3B*, *Figure 3—figure supplement 1A*).

We confirmed that complex I or complex III inhibition elevated whole-cell $NADH/NAD^+$, which primarily reflects the mitochondrial ratio (*Titov et al., 2016*; *Sies, 1982*; *Eng et al., 1989*), more so than ATP synthase inhibition (*Figure 3C*), consistent with leak respiration in the latter case. The increase in whole-cell $NADH/NAD^+$ was prevented in all cases by mito*Lb*NOX, as expected, whereas *Lb*NOX provided only limited relief given the smaller cytosolic contribution to the whole-cell signal.

We also measured the secreted [lactate]/[pyruvate] ratio, a proxy for cytosolic $NADH/NAD^+$ through equilibration by the lactate dehydrogenase (LDH) reaction (*Williamson et al., 1967*; *Figure 3D*). The results aligned with the predicted effects of the inhibitors on the malate-aspartate shuttle and the glycerol-3-phosphate shuttle, which facilitate oxidation of cytosolic reducing equivalents in mitochondria (*Dawson, 1979*; *Safer et al., 1971*). Thus, the increase in mitochondrial $NADH/NAD^+$ in either piericidin- or antimycin-treated cells is expected to stall reactions coupled to this ratio in the tricarboxylic acid (TCA) cycle that are required for malate-aspartate shuttle activity (*LaNoue and Williamson, 1971*). However, only antimycin overly-reduces the CoQ pool, which blocks the TCA cycle at complex II independently of $NADH/NAD^+$ and additionally inhibits the glycerol-3-phosphate shuttle (*Mráček et al., 2013*; *Figure 3—figure supplement 1B*).

LbNOX expression was able to maintain a low [lactate]/[pyruvate] ratio in all cases, as it directly oxidizes the cytosol (*Figure 3D*). mito*Lb*NOX also buffered the impact of piericidin and oligomycin but failed to effectively oxidize the cytosol during antimycin treatment. This result demonstrated that mito*Lb*NOX affected cytosolic $NADH/NAD^+$ through the redox shuttles, both of which are inhibited by antimycin's effect on CoQ in a manner that mito*Lb*NOX cannot resolve.

Strikingly, *Lb*NOX expression was sufficient to almost completely ablate ISR activation, represented by the *Ddit3* transcript, upon complex I or complex III inhibition but only partially mitigated ISR activation by ATP synthase inhibition (*Figure 3E*). mito*Lb*NOX also significantly attenuated ISR activation during complex I inhibition but proved less effective during complex III inhibition, possibly due to its failure to fully oxidize the cytosol in this case.

We performed RNA-sequencing to ascertain the global effects of maintaining compartment-specific NADH oxidation during inhibitor treatments. PCA of the combined data from control myoblasts (shown in *Figure 2*) and myoblasts expressing *Lb*NOX, mito*Lb*NOX or NDI1 recapitulated a first principal component driven by the ISR and proliferative gene expression. The effects following

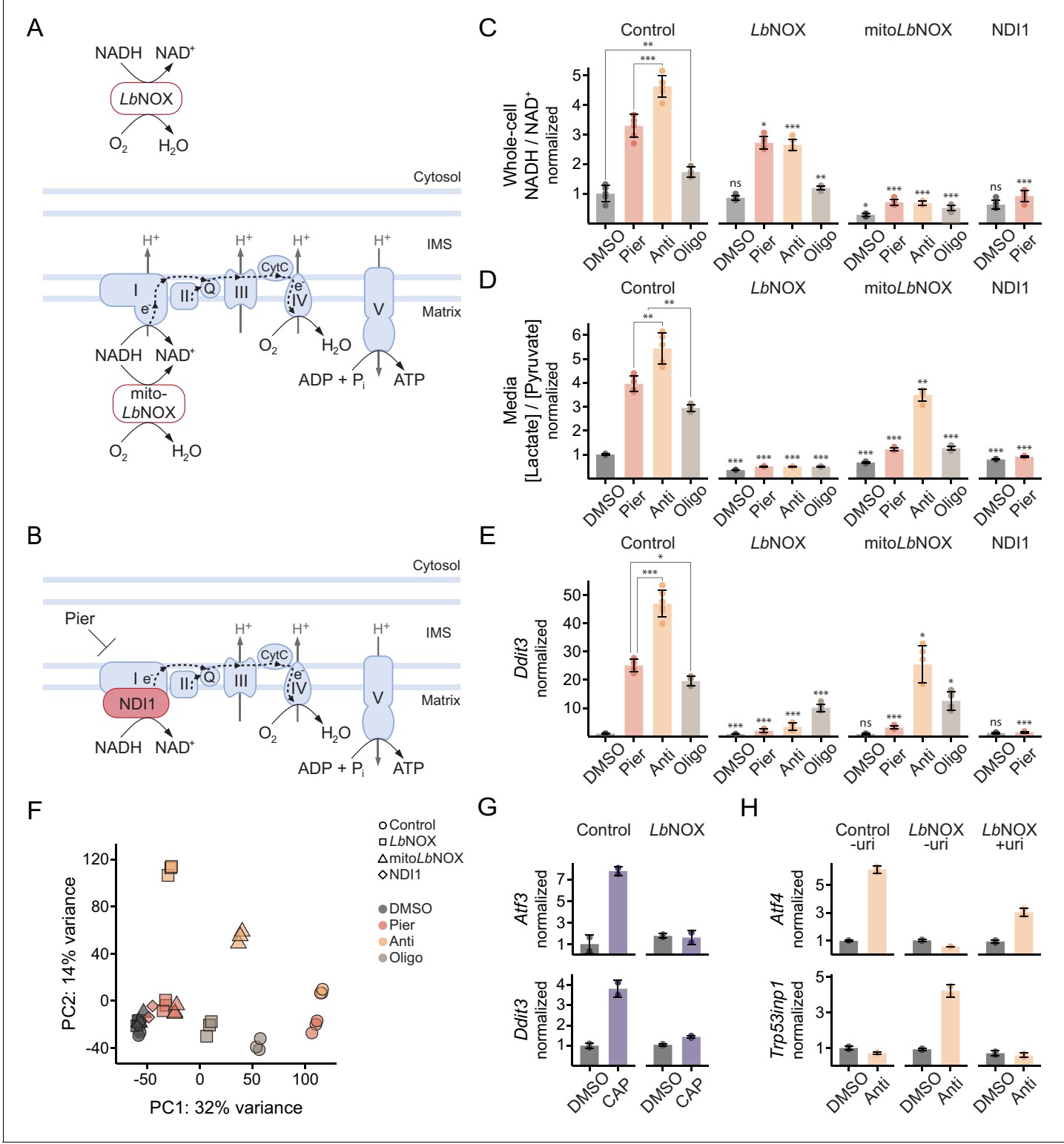

**Figure 3.** Oxidizing cytosolic NADH/NAD$^+$ is sufficient to ablate ISR activation by complex I inhibition in myoblasts. (A) Schematic of localization and activity of *Lb*NOX and mito*Lb*NOX. IMS, intermembrane space. (B) Schematic of localization and activity of NDI1. (C) NADH/NAD$^+$ in extracts of cells expressing luciferase (control), *Lb*NOX, mito*Lb*NOX or NDI1 and treated with inhibitors for 1 hr. Data is normalized to DMSO in control cells. Mean ± SD, N = 5–6 from two experiments. Welch's t-test (two-tailed) was used to compare each pair of treatments within control cells, as well as each treatment in the other cells with its equivalent in control cells (significance notations with no connecting lines), followed by Holm's correction for multiple testing. (D) Media [lactate]/[pyruvate] following 2 hr treatments in control, *Lb*NOX, mito*Lb*NOX or NDI1 cells. Data is normalized to DMSO in

*Figure 3 continued on next page*

*Figure 3 continued*

control cells. Mean ± SD, N = 5–6 from three experiments. Statistical analysis as in C. (E) qPCR of *Ddit3* following 10 hr treatments in control, *Lb*NOX, mito*Lb*NOX or NDI1 cells. Data is presented as fold-change from DMSO in control cells. Mean ± SD, N = 5–6 from two experiments. Statistical analysis as in C on the ΔΔC$_t$ values. (F) PCA of gene expression levels derived from RNA-seq in control, *Lb*NOX, mito*Lb*NOX or NDI1 cells following 10 hr treatments, as indicated. Detailed results are provided in *Supplementary files 1* and *2*. (G) Fold-change from DMSO derived from RNA-seq for *Atf3* and *Ddit3* in control and *Lb*NOX cells treated for 48 hr with chloramphenicol (CAP) in the presence of uridine. Mean ± SD, N = 2. (H) qPCR of *Atf4* and *Trp53inp1* in control and *Lb*NOX cells following 10 hr antimycin treatment, with or without uridine (uri). Data is presented as fold-change from DMSO in control cells without uridine. Mean ± SD, N = 2. ns, p>0.05; *, p<0.05; **, p<0.01; ***, p<0.001.

The online version of this article includes the following figure supplement(s) for figure 3:

**Figure supplement 1.** Additional data on bioenergetics and metabolism in myoblast cell lines.

inhibitor treatments were broadly in line with the results for *Ddit3*. During complex I inhibition (red), oxidizing the cytosol with *Lb*NOX (square), or the matrix and the cytosol with mito*Lb*NOX (triangle), was almost as effective in preventing gene expression changes along PC1 as fully maintaining OXPHOS with NDI1 (diamond) (*Figure 3F*; *Supplementary file 1*). Thus, impaired NADH oxidation was the primary trigger of ISR-related gene expression in myoblasts and correcting the NADH imbalance in the cytosol was sufficient to prevent this, which was also evident following 48 hr of inhibiting mitochondrial protein synthesis with chloramphenicol (*Figure 3G*).

## Oxidizing cytosolic NADH/NAD$^+$ during complex III inhibition in myoblasts promotes a p53 response that ablates the ISR

Inhibition of complex III, like complex I, leads to an increased NADH/NAD$^+$ ratio, and oxidizing the cytosol using *Lb*NOX similarly negated gene expression changes along PC1 during complex III inhibition (yellow; *Figure 3F*). However, it also triggered an orthogonal signature along PC2. *Cis*-regulatory analysis of the 500 genes with the most positive PC2 weights highlighted p53 as a possible regulator of this signature (Materials and methods; *Supplementary file 2*). Indeed, canonical p53 effectors such as *Cdkn1a* and *Trp53inp1* were among the top 50 genes.

Complex III dysfunction has been shown to activate p53 due to a pyrimidine deficiency that results from inability of dihydroorotate dehydrogenase (DHODH) to donate electrons to CoQ (*Khutornenko et al., 2010*). p53 activation downregulated *Atf4* in this setting and shut down ISR gene expression (*Evstafieva et al., 2014*). Given these observations, we compared the *Atf4* and *Trp53inp1* transcripts following antimycin treatment. As before, control cells activated the ISR but not p53 while *Lb*NOX cells activated a p53 response that inhibited *Atf4* (*Figure 3H*). p53 activation in *Lb*NOX cells was prevented by adding uridine to compensate for the pyrimidine deficiency, in which case ISR gene expression was not fully ablated. Thus, p53 activation following complex III inhibition was dependent on an oxidized cytosolic NADH/NAD$^+$ ratio and required to fully ablate the ISR.

## Complex I inhibition in myoblasts activates the eIF2α kinase GCN2 due to an asparagine deficiency

We focused our mechanistic investigation on the functional consequence of complex I inhibition that triggered the ISR, as the response was most effectively blunted by oxidizing cytosolic NADH in this case. One possibility was that glycolysis was inhibited by elevated cytosolic NADH/NAD$^+$, limiting the cells' ability to defend their adenylate energy charge in the absence of OXPHOS. Using lactate secretion as a proxy for glycolytic flux, we found that *Lb*NOX did modestly stimulate glycolysis (*Figure 4A*, *Figure 4—figure supplement 1A*). However, this did not translate into an improved energy charge, and in fact, the opposite was true (*Figure 4B*, *Figure 4—figure supplement 1B*).

Another possibility was that elevated mitochondrial and cytosolic NADH/NAD$^+$ depleted a critical nutrient, which then triggered the ISR. We performed intracellular metabolite profiling on control cells or *Lb*NOX cells acutely treated with piericidin (*Figure 4C*, *Figure 4—figure supplement 1C*). The amino acids aspartate and its derivative asparagine, alongside related TCA cycle intermediates, emerged among the top metabolites depleted by piericidin in a manner responsive to oxidizing the cytosol. This result is consistent with the contribution to aspartate synthesis of the mitochondrial and cytosolic NAD$^+$-linked malate dehydrogenase (MDH) reactions that yield its precursor, oxaloacetate

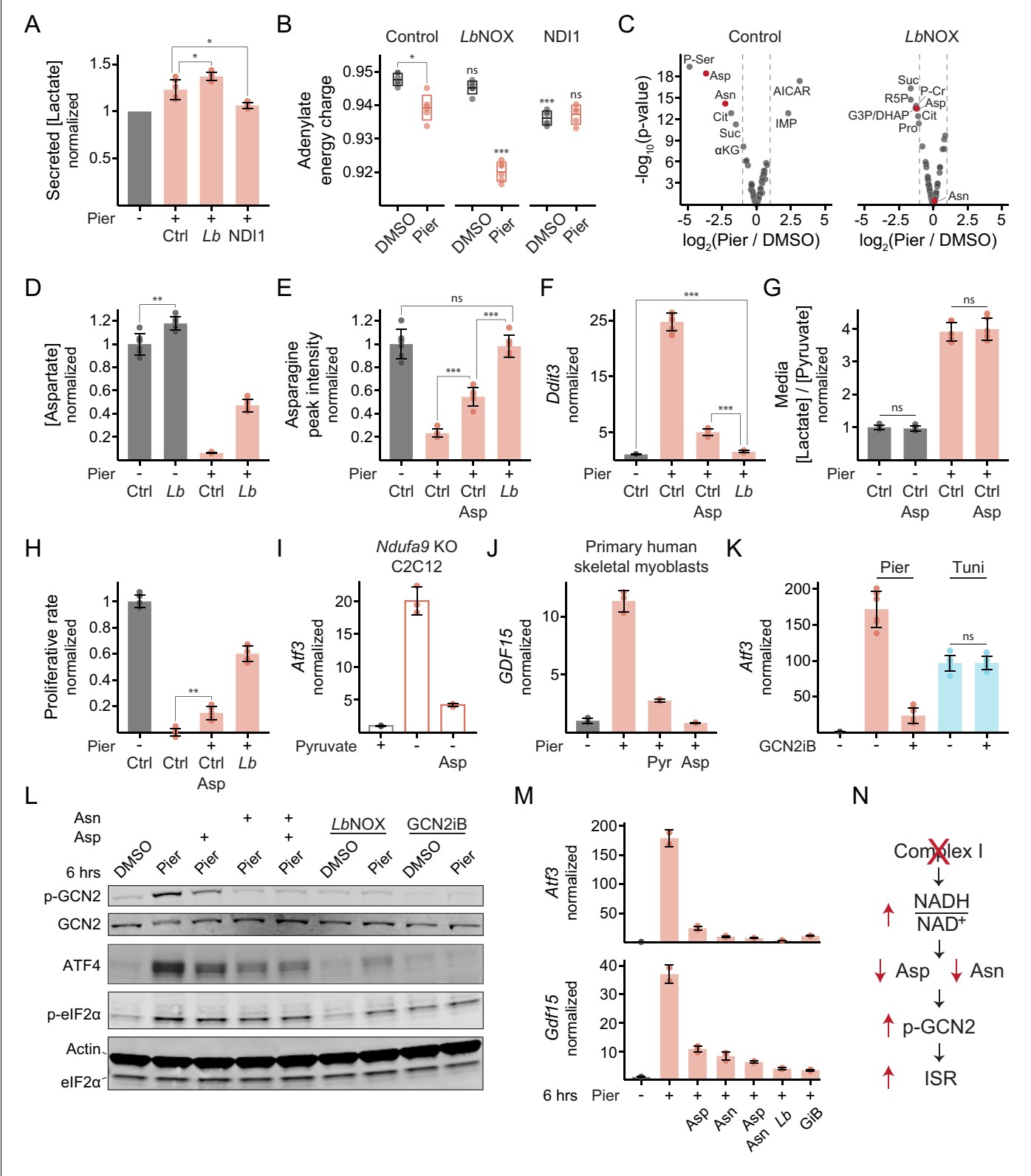

**Figure 4.** Complex I inhibition in myoblasts triggers the ISR through the kinase GCN2 due to an aspartate and asparagine deficiency. (**A**) Secreted [lactate] following 2 hr piericidin treatment in control (Ctrl), *Lb*NOX (*Lb*) or NDI1 cells. Data is normalized to DMSO (-) separately for each cell line. Mean ± SD, N = 6 from three experiments (same samples as in *Figure 3D*). Welch's t-test (two-tailed) was used to compare control with *Lb*NOX and NDI1 cells, followed by Holm's correction for multiple testing. (**B**) Adenylate energy charge following 1 hr piericidin treatment in control, *Lb*NOX or

*Figure 4 continued on next page*

*Figure 4 continued*

NDI1 cells. Mean ± SD, N = 5–6 from two experiments. The Games-Howell test was used to make all pairwise comparisons. Notations with no connecting lines relate to the equivalent treatment in control cells. (C) Fold-change (x-axis) and statistical significance (y-axis) of metabolite differential abundance following 1 hr piericidin treatment in extracts of control or *Lb*NOX cells. N = 3. P-Ser, phosphoserine; Asp, aspartate; Asn, asparagine; Cit, citrate; Suc, succinate; αKG, α-ketoglutarate; IMP, inosine monophosphate; AICAR, 5-aminoimidazole-4-carboxamide ribonucleotide; R5P, ribose 5-phosphate; G3P/DHAP, glyceraldehyde 3-phosphate and/or dihydroxyacetone phosphate; P-Cr, phosphocreatine; Pro, proline. Detailed results are provided in *Figure 4—source data 1*. (D) Intracellular [aspartate] following 1 hr piericidin treatment in control or *Lb*NOX cells. Data is normalized to DMSO in control cells. Mean ± SD, N = 6 from two experiments (includes samples shown in C). The Games-Howell test was used for all pairwise comparisons. (E) Peak intensity of intracellular asparagine following 1 hr piericidin treatment in control cells, with or without aspartate, and in *Lb*NOX cells. Data is normalized to DMSO in control cells. Mean ± SD, N = 6 from two experiments. The Games-Howell test was used for all pairwise comparisons. (F) qPCR of *Ddit3* following 10 hr piericidin treatment in control cells, with or without aspartate, and in *Lb*NOX cells. Data is presented as fold-change from DMSO in control cells. Mean ± SD, N = 8 from three experiments. The Games-Howell test was used to make all pairwise comparisons of ΔΔC$_t$ values. (G) Media [lactate]/[pyruvate] following 2 hr piericidin treatment, with or without aspartate, in control cells. Data is normalized to DMSO without aspartate. Mean ± SD, N = 6 from three experiments. Welch's t-test (two-tailed) was used to compare each treatment with and without aspartate, followed by Holm's correction. (H) Proliferative rate (doublings in 24 hr) of control cells, with or without aspartate, and of *Lb*NOX cells following piericidin treatment. Data is normalized to DMSO in control cells. Mean ± SD, N = 5–6 from three experiments. The Games-Howell test was used for all pairwise comparisons. (I) qPCR of *Atf3* following 10 hr pyruvate withdrawal, with or without aspartate, in *Ndufa9*-KO C2C12 myoblasts. Data is presented as fold-change from the condition with pyruvate (+). Mean ± SD, N = 3. (J) qPCR of *GDF15* following 10 hr piericidin treatment, with or without pyruvate or aspartate, in primary human skeletal myoblasts. Data is presented as fold-change from DMSO. Mean ± SD, N = 3. (K) qPCR of *Atf3* following 10 hr piericidin or tunicamycin (Tuni) treatment, with or without GCN2iB, in control cells. Data is presented as fold-change from DMSO. Mean ± SD, N = 6-7. Welch's t-test (two-tailed) was used to compare each treatment with and without GCN2iB, followed by Holm's correction. (L) Western blot of (p-)GCN2, ATF4 and (p-)eIF2α following 6 hr piericidin treatment in the indicated conditions in *Lb*NOX cells. *Lb*NOX expression was induced only where indicated. (M) qPCR of *Atf3* and *Gdf15* in the same cells and conditions shown in L. Data is presented as fold-change from DMSO. Mean ± SD, N = 2–3. GiB, GCN2iB. (N) Model for ISR activation by complex I inhibition in myoblasts. ns, p>0.05; *, p<0.05; **, p<0.01; ***, p<0.001. The online version of this article includes the following source data and figure supplement(s) for figure 4:

**Source data 1.** Metabolite profiling data.
**Figure supplement 1.** Additional data on metabolic consequences that trigger the ISR in myoblasts.

---

(*Birsoy et al., 2015*; *Sullivan et al., 2015*; *Chen et al., 2016*). Absolute quantification confirmed a ~ 16 fold drop in aspartate within 1 hr of piericidin treatment in control cells compared with only a 2.5-fold drop in *Lb*NOX cells (*Figure 4D*, *Figure 4—figure supplement 1D*).

Amino acid deficiency is a canonical ISR trigger through the eIF2α kinase GCN2, which is activated due to uncharged tRNA (*Hinnebusch, 1984*; *Dever et al., 1992*; *Wek et al., 1995*; *Berlanga et al., 1999*; *Zhang et al., 2002*; *Castilho et al., 2014*; *Inglis et al., 2019*; *Harding et al., 2019*). If amino acid deficiency triggers the ISR following complex I inhibition in myoblasts, then correcting the deficiency should abrogate the response. We began by adding aspartate since the magnitude of its depletion was greater and its addition also allows the cells to partially restore asparagine (*Figure 4E*, *Figure 4—figure supplement 1E*). Indeed, 10 mM aspartate significantly attenuated ISR activation upon complex I inhibition, though not to the same extent as *Lb*NOX (*Figure 4F*, *Figure 4—figure supplement 1F–I*). This concentration was required since most mammalian cells do not efficiently take up aspartate (*Birsoy et al., 2015*). Importantly, aspartate addition did not indirectly oxidize cytosolic NADH/NAD⁺ (*Figure 4G*) and was insufficient to appreciably restore cell proliferation (*Figure 4H*). Aspartate also attenuated ISR-related gene expression in C2C12 myoblasts engineered with a genetic defect in complex I (*Vafai et al., 2016*), where reductive stress was induced by pyruvate withdrawal (*Figure 4I*), as well as in piericidin-treated primary human myoblasts (*Figure 4J*) and primary mouse embryonic fibroblasts (*Figure 4—figure supplement 1J*).

To test whether ISR activation by complex I inhibition involved GCN2 activity, we used a recently described specific, small-molecule, ATP-competitive inhibitor of the kinase (GCN2iB) (*Nakamura et al., 2018*). Indeed, co-treatment with GCN2iB significantly suppressed ISR gene expression in response to complex I inhibition but had no effect on ISR activation in response to endoplasmic reticulum (ER) stress induced by tunicamycin (*Figure 4K*). The converse was true when we inhibited the ER-resident eIF2α kinase, PERK (*Figure 4—figure supplement 1K*; *Harding et al., 1999*; *Axten et al., 2013*).

Finally, we wished to examine how GCN2 sensed the amino acid deficiency. Using autophosphorylation at Threonine-898 as a readout of kinase activation (*Romano et al., 1998*), we observed that 6

hr of complex I inhibition resulted in marked GCN2 activation, which was partially attenuated by supplementing aspartate (*Figure 4L*). Strikingly, asparagine alone abrogated GCN2 activation with no discernible additive effect of aspartate, suggesting the drop in asparagine was the most proximal activating signal. As expected, *Lb*NOX expression, which enables synthesis of both amino acids during complex I inhibition, effectively prevented GCN2 activation, while GCN2iB suppressed autophosphorylation both at baseline and upon piericidin treatment.

The effects of the same interventions on eIF2α phosphorylation, the event downstream of GCN2 activation, were notably less pronounced, especially given the narrow dynamic range of this signal. We observed clear attenuation of eIF2α phosphorylation upon piericidin treatment in the case of *Lb*NOX expression, though the rescue was still incomplete (*Figure 4L*, *Figure 4—figure supplement 1L*). ATF4 protein levels largely tracked the degree of GCN2 activation (*Figure 4L*) and this was reflected at the level of transcriptional ISR targets at the same time-point (*Figure 4M*). GCN2iB completely prevented ATF4 protein accumulation despite some residual eIF2α phosphorylation (*Figure 4L*), suggesting GCN2 was required to attain the threshold phosphorylation level that elicits ATF4 translation. Collectively, these experiments indicate that in myoblasts, the rise in the NADH/NAD$^+$ ratio is a major driver of ISR-related gene expression following complex I inhibition, as it limits biosynthesis of aspartate, depletes asparagine and activates GCN2 (*Figure 4N*).

## ATP synthase inhibition potently triggers the ISR in myotubes while ETC inhibition neither depletes asparagine nor activates the response

We next examined the effects of inhibitor treatments in myotubes. In contrast to myoblasts, myotubes only experienced significant gene expression changes upon ATP synthase inhibition (PC1; 50% of the variance) (*Figure 5A*; *Supplementary file 1*), driven once more by ISR activation (*Figure 5B*), whereas ETC inhibition had little effect. The shift in the pattern of ISR activation was already evident as soon as 24 hr after the switch to low serum, when cells are largely post-mitotic but far from differentiated (*Figure 5—figure supplement 1A*).

To interrogate why myotubes did not trigger the ISR upon ETC inhibition, we again measured compartment-specific NADH/NAD$^+$. Myotubes did experience a significant increase in both the mitochondrial and the cytosolic ratio during ETC inhibition (*Figure 5C,D*). However, the ratios were not tightly linked, as in myoblasts. The severity of NADH imbalance across the inhibitors was discordant between the compartments and oxidizing mitochondrial NADH with mito*Lb*NOX failed to oxidize cytosolic NADH/NAD$^+$. These results suggest myotubes did not rely on complex I to oxidize cytosolic reducing equivalents via the malate-aspartate shuttle. Rather, cytosolic NADH/NAD$^+$ likely increased due to activation of glycolysis, evidenced by a ~ 75% rise in lactate secretion (*Figure 5E*), that was triggered by the drop in the energy charge regardless of the specific inhibitor used (*Figure 5F*). The glycerol-3-phosphate shuttle, which is uniquely sensitive to antimycin, remained operative (*Figure 5D*).

We next wondered how the altered metabolic state of myotubes impacted the fate of the amino acids aspartate and asparagine, whose depletion triggered the ISR in myoblasts. Despite elevated mitochondrial and cytosolic NADH/NAD$^+$, complex I or complex III inhibition caused only a modest decrease (2-fold and 4-fold, respectively) in intracellular aspartate whereas ATP synthase inhibition even led to a slight accumulation (*Figure 5G*). None of the treatments depleted asparagine, which in fact accumulated (*Figure 5H*), possibly reflecting elevated cytosolic aspartate as asparagine synthesis occurs in the cytosol. The absence of asparagine deficiency likely explains why myotubes do not activate the ISR when complex I or complex III is inhibited. Myotubes also failed to trigger the ISR after mitochondrial protein synthesis was blocked for 48 hr (*Figure 5—figure supplement 1B*).

## ISR activation by ATP synthase inhibition in myotubes is related to mitochondrial inner-membrane hyperpolarization

We next explored why ATP synthase inhibition activates the ISR in myotubes. Inhibition of complex I, complex III or ATP synthase all acutely impair NADH oxidation and OXPHOS but their effects on mitochondrial membrane potential are distinct. Only ATP synthase inhibition is associated with inner-membrane hyperpolarization. To test whether this phenomenon was responsible for ISR activation, we treated myotubes with oligomycin in combination with piericidin, which prevents hyperpolarization by eliminating residual ETC activity and proton pumping. We also treated with oligomycin

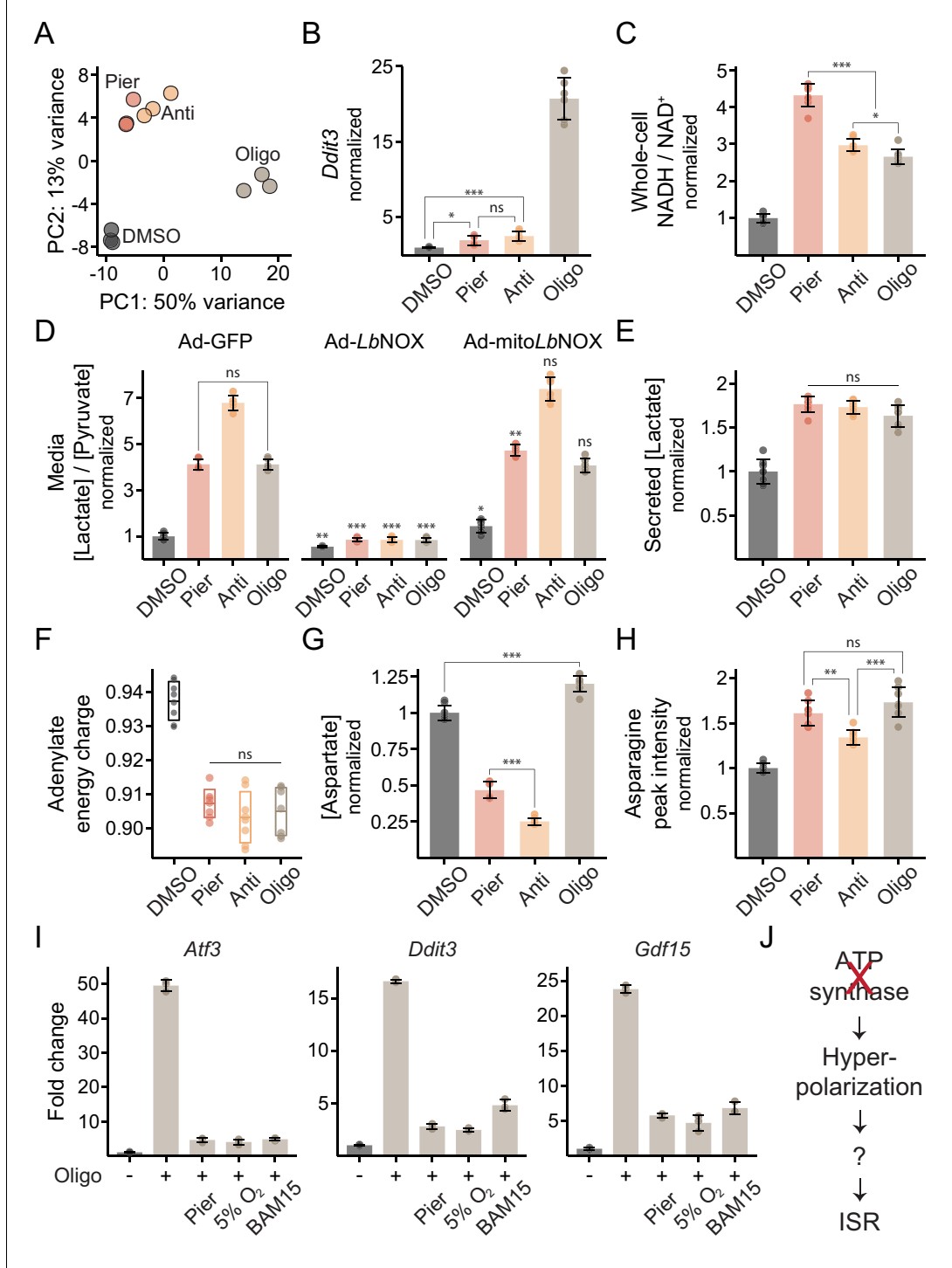

**Figure 5.** ATP synthase inhibition triggers the ISR in myotubes in a manner related to inner-membrane hyperpolarization. (A) PCA of gene expression levels derived from RNA-seq in myotubes treated with inhibitors for 10 hr. N = 3 (same samples as in *Figure 1D*). Detailed results are provided in *Supplementary file 1*. (B) qPCR of *Ddit3* following 10 hr treatments in myotubes. Data is presented as fold-change from DMSO. Mean ± SD, N = 6 from two experiments. The Games-Howell test was used for all pairwise comparisons of $\Delta\Delta C_t$ values. (C) NADH/NAD$^+$ in myotube extracts following 1 hr treatments. Data is normalized to DMSO. Mean ± SD, N = 8 from two experiments. The Games-Howell test was used for all pairwise comparisons. (D) Media [lactate]/[pyruvate] following 2 hr treatments in myotubes expressing GFP, *Lb*NOX or mito*Lb*NOX using adenoviral transduction. Data is normalized to DMSO in GFP. Mean ± SD, N = 6 from three experiments. Welch's t-test (two-tailed) was used to compare each pair of treatments within GFP cells, as well as each treatment in the other cells with its equivalent in GFP cells (significance notations with no connecting lines), followed by Holm's correction for multiple testing. (E) Secreted [lactate] following 2 hr treatments in myotubes. Data is normalized to DMSO. Mean ± SD, N = 6

*Figure 5 continued on next page*

*Figure 5 continued*

from three experiments (same samples as in D). (F) Adenylate energy charge following 1 hr treatments in myotubes. Mean ± SD, N = 7–8 from two experiments. The Games-Howell test was used for all pairwise comparisons. (G) Intracellular [aspartate] following 1 hr treatments in myotubes. Data is normalized to DMSO. Mean ± SD, N = 7–8 from two experiments. The Games-Howell test was used for all pairwise comparisons. (H) Peak intensity of intracellular asparagine following 1 hr treatments in myotubes. Data is normalized to DMSO. Mean ± SD, N = 8 from two experiments. The Games-Howell test was used for all pairwise comparisons. (I) Fold-change from DMSO derived from RNA-seq for *Atf3*, *Ddit3* and *Gdf15* in myotubes treated for 10 hr with oligomycin alone, or in combination with piericidin, BAM15 or 5% $O_2$. Mean ± SD, N = 3. See also *Supplementary file 1*. (J) Model for ISR activation by ATP synthase inhibition in myotubes. ns, p>0.05; *, p<0.05; **, p<0.01; ***, p<0.001.

The online version of this article includes the following figure supplement(s) for figure 5:

**Figure supplement 1.** Additional data on ISR activation in myotubes.

after pre-conditioning in mild hypoxia (5% $O_2$), which dampens residual ETC activity by shunting pyruvate flux away from the TCA cycle (*Papandreou et al., 2006*; *Wheaton and Chandel, 2011*; *Semenza, 2011*). Alternatively, we treated with oligomycin in combination with a mild dose of BAM15 to directly depolarize the inner-membrane (*Kenwood et al., 2014*). All three interventions significantly attenuated the ISR (*Figure 5I*, *Figure 5—figure supplement 1C*; *Supplementary file 1*), whereas *Lb*NOX expression did not (*Figure 5—figure supplement 1D*; *Supplementary file 1*), implicating hyperpolarization as the key factor in the signal relay (*Figure 5J*). Similar results were obtained in primary human myotubes (*Figure 5—figure supplement 1E*). Intriguingly, a higher dose of BAM15 triggered the ISR on its own, suggesting perturbation to membrane potential in either direction was sensed (*Figure 5—figure supplement 1C*).

## Discussion

Several recent studies have reported activation of the integrated stress response (ISR) during mitochondrial dysfunction, however, tracing the underlying mechanisms has proved challenging. Here, we interrogated this response in two cell states, proliferating myoblasts and differentiated myotubes, by employing a battery of small-molecule inhibitors of mitochondria in combination with genetic and chemical tools for selectively buffering their effects. A chief result of this work is that there is no universal mechanism linking mitochondrial dysfunction to ISR activation but, rather, manifold paths that depend on the nature of the mitochondrial defect and on the metabolic state of the cell.

Our results in proliferating myoblasts mechanistically relate two recently recognized consequences of ETC dysfunction: a defect in aspartate biosynthesis and ISR activation. Aspartate is largely produced from TCA cycle-derived oxaloacetate (OAA) but elevated mitochondrial NADH/NAD$^+$ blocks the oxidative TCA cycle when complex I is inhibited (*Chen et al., 2016*). OAA must then be obtained by other means, which in myoblasts depended on correcting the concomitant rise in cytosolic NADH/NAD$^+$, thus tilting the MDH1 reaction toward OAA (*Birsoy et al., 2015*; *Sullivan et al., 2015*). Absent this, we found the aspartate deficiency triggered the ISR, at least initially due to depletion of its derivative asparagine that was sensed by the eIF2α kinase GCN2 (*Figure 4C–M*). The cytosolic aspartate and asparagine biosynthesis enzymes are both ISR targets (*Fujita et al., 2007*; *Harding et al., 2003*; *Han et al., 2013*), suggesting a homeostatic logic to the response.

Supplementation with asparagine alone, which cannot be converted to aspartate in most mammalian cells (*Pavlova et al., 2018*), was sufficient to abrogate GCN2 activation (*Figure 4L*). That GCN2 did not directly sense the profound aspartate depletion seems surprising. This result may be explained by the fact that while the $K_M$ values for aspartate and asparagine of their respective cytosolic tRNA synthetases are similar (~15–30 μM) (*Bour et al., 2009*; *Messmer et al., 2009*; *Andrulis et al., 1978*), the cellular concentrations of these amino acids are orders of magnitude apart. Aspartate is estimated at ~5–10 mM whereas asparagine is much closer to the $K_M$, at ~100–200 μM (*Chen et al., 2016*; *Park et al., 2016*). Thus, even a large fold-change drop in aspartate would not compromise its cognate tRNA charging but a smaller drop in asparagine would, activating GCN2. The $K_M$ for aspartate of asparagine synthetase is much higher (~1 mM) (*Horowitz and Meister, 1972*; *Ciustea et al., 2005*), so aspartate deficiency readily depletes asparagine (*Figure 4E*). We conclude asparagine tRNA charging is a sensitive sensor of NADH reductive stress in proliferating cells.

We note that oxidizing cytosolic NADH/NAD$^+$ using *Lb*NOX attenuated the ISR better than aspartate or asparagine (*Figure 4F,L,M*), suggesting eIF2$\alpha$ kinases may be attuned to additional consequences of reductive stress. Moreover, *Lb*NOX cells still showed residual ISR activation (*Figure 4L*, *Figure 4—figure supplement 1L*), likely due to reasons beyond reductive stress. However, our results suggest threshold and non-linear effects along the ISR pathway so that NADH reductive stress was the primary driver of ISR-related gene expression.

Surprisingly, we also found that NADH reductive stress was a key modulator of an interaction between the ISR and p53 during complex III inhibition in myoblasts. The pyrimidine deficiency resulting from an overly-reduced CoQ triggered p53 to shut down ISR gene expression only when cytosolic reductive stress was resolved (*Figure 3G*). It is likely that the break on DNA replication imposed by reductive stress prevented the pyrimidine deficiency from giving rise to DNA damage, so p53 was not activated. Alternatively, cytosolic NADH/NAD$^+$ may regulate p53 itself, possibly via NAD(P)H:quinone oxidoreductase 1 (NQO1) (*Khutornenko et al., 2010*; *Asher et al., 2001*; *Asher et al., 2002*), which could have implications for cancer biology.

The mapping between mitochondrial perturbation and ISR activation changed in non-dividing cells (*Figure 1D*, *Figure 5B*, *Figure 5—figure supplement 1A*). The ISR was not triggered by ETC inhibition in myotubes (*Figure 5B*), in line with a much smaller drop in aspartate and no drop in asparagine (*Figure 5G,H*). This result could be explained by reduced demand for these amino acids for protein and nucleotide biosynthesis (*Clissold and Cole, 1973*; *Krauter et al., 1979*; *Bowman, 1987*; *Larson et al., 1993*; *Zahradka et al., 1991*; *Frangini et al., 2013*; *Talvas et al., 2006*), and by increased capacity for amino acid turnover or scavenging (*Deval et al., 2008*; *Sadiq et al., 2007*), in myotubes relative to myoblasts. Moreover, myoblasts relied on complex I to oxidize cytosolic reducing equivalents via the malate-aspartate shuttle whereas this was absent in myotubes (*Figure 3D*, *Figure 5D*), consistent with observations in skeletal muscle in vivo (*Sahlin et al., 1987*; *MacDonald and Marshall, 2000*; *Alfadda et al., 2004*). Myotubes did exhibit elevated cytosolic NADH/NAD$^+$ upon complex I inhibition but this was likely due to activation of glycolysis (*Figure 5E*). Conversion of glucose to lactate is net redox neutral at steady-state but elevated NADH/NAD$^+$ can emerge when flux through glyceraldehyde 3-phosphate dehydrogenase (GAPDH) acutely rises or through siphoning of intermediates between GAPDH and LDH (*Noor et al., 2013*; *Shestov et al., 2014*).

Myotubes activated the ISR upon ATP synthase inhibition (*Figure 5B*), independently of a drop in ATP (*Figure 5F*) or impaired NADH oxidation (*Figure 5C,D*). The response was attenuated by eliminating residual respiration or by mild depolarization (*Figure 5I*, *Figure 5—figure supplement 1C,E*), implicating membrane potential in the signal relay. In respiring cells, oligomycin typically hyperpolarizes the inner-membrane as it blocks a major consumer of the proton gradient (*Brand and Nicholls, 2011*). Eliminating residual respiration relieves hyperpolarization by preventing proton pumping while depolarization involves facilitated proton re-entry into the matrix. This result is reminiscent of another case we recently described where inhibition of complex I ameliorated proliferative and metabolic defects due to ATP synthase inhibition, in part by relieving hyperpolarization (*To et al., 2019*). The link to hyperpolarization may explain why this route to the ISR is more prominent in myotubes, which exhibit increased and tightly coupled OXPHOS, though it likely plays a role in oligomycin-treated myoblasts as well. We note that severe depolarization also triggered the ISR in myotubes (*Figure 5—figure supplement 1C*).

While this paper was in revision, Guo et al. and Fessler et al. reported that the eIF2$\alpha$ kinase HRI potently triggers the ISR in response to either ATP synthase inhibition or severe depolarization (*Fessler et al., 2020*; *Guo et al., 2020*), in agreement with a previous study in the latter case (*Taniuchi et al., 2016*). HRI activation depended on processing of the inner-membrane protein DELE1 by the inner-membrane protease OMA1 (*Fessler et al., 2020*; *Guo et al., 2020*). It is tempting to speculate this mechanism is operative in our system, as OMA1 has been shown to respond to both depolarization and hyperpolarization (*Ehses et al., 2009*; *Baker et al., 2014*; *Zhang et al., 2014*), consistent with our observation that perturbation of membrane potential in either direction triggers the response. How such opposite effects converge in terms of signaling is unclear but our results suggest the common denominator is not, as Fessler et al. proposed, the drop in ATP (*Figure 5F,I*, *Figure 5—figure supplement 1C,E*).

Our results exclude mitochondrial processes that do not seem to be involved in ISR activation. The response was not activated simply due to the effects of OXPHOS inhibition on energy

metabolism, so long as glucose was available. Similarly, we did not observe contributions to ISR activation by metabolic consequences exclusively related to elevated intra-mitochondrial NADH/NAD$^+$. And blockade of mtDNA-encoded protein synthesis did not appear to be directly monitored in either cell state (*Figure 3G*, *Figure 5—figure supplement 1B*). The advantage of our cellular system is that so many parameters can be carefully controlled, however, it has important limitations: (i) chemical ETC inhibition is an imperfect model for genetic defects, which may impact multiple complexes and act over longer time scales, and (ii) cell culture conditions represent an artificial nutrient milieu, do not model interactions with other cell types and do not subject the cells to physiological workloads.

Our study demonstrates that multiple paths connect mitochondrial dysfunction to the ISR, depending on the combination of the defect and the metabolic state of the cell. This framework may prove useful in explaining variability in ISR activation across cell types and stressors and in dissecting the logic of the ISR in mitochondrial disorders. There is controversy over whether the ISR is activated in vivo directly by mtDNA expression defects or by ETC/OXPHOS defects (*Kühl et al., 2017*; *Dogan et al., 2014*; *Lehtonen et al., 2016*). Our results help address it by defining mechanisms through which the latter trigger the response. We note pyruvate therapy, which serves to regenerate NAD$^+$, was recently shown to decrease serum GDF15, a circulating marker of the ISR, in mitochondrial disease patients (*Koga et al., 2019*), providing rich in vivo support for our identification of NADH imbalance as a trigger of the ISR. Further study of ISR activation and its consequences in the context of mitochondrial dysfunction may help define ways to therapeutically modulate the response.

# Materials and methods

## Key resources table

| Reagent type (species) or resource | Designation | Source or reference | Identifiers | Additional information |
|---|---|---|---|---|
| Cell line (*Mus musculus*) | C2C12 | ATCC | CRL-1772 | |
| Cell line (*M. musculus*) | *Ndufa9*-knockout C2C12 | PMID:27622560 | | |
| Cell line (*M. musculus*) | Primary mouse embryonic fibroblasts (MEF) | Lonza | M-FB-481 | |
| Cell line (*Homo sapiens*) | Primary human skeletal myoblasts (HSkM) | Thermo Fisher Scientific | A11440 | |
| Recombinant DNA reagent | Doxycycline-inducible luciferase (pLVX-TRE3G-Luc) | Takara Bio (formerly Clontech) | Sold as part of 631187 | |
| Recombinant DNA reagent | Doxycycline-inducible *Lb*NOX | PMID:27124460 | | Available without inducible system on Addgene (#75285) |
| Recombinant DNA reagent | Doxycycline-inducible mito*Lb*NOX | PMID:27124460 | | Available without inducible system on Addgene (#74448) |
| Recombinant DNA reagent | Doxycycline-inducible *Saccharomyces cerevisiae* NDI1 | PMID:27124460 | | |
| Other | Human adenovirus type5 expressing eGFP | Vector Biolabs | 1060-HT | |
| Other | Human adenovirus type5 expressing eGFP and *Lb*NOX | Vector Biolabs (PMID:27124460) | | |
| Other | Human adenovirus type5 expressing eGFP and mito*Lb*NOX | Vector Biolabs (PMID:27124460) | | |

*Continued on next page*

*Continued*

| Reagent type (species) or resource | Designation | Source or reference | Identifiers | Additional information |
|---|---|---|---|---|
| Other | Dulbecco's Modified Eagle's Medium (DMEM), 4 mM *L*-Glutamine, no glucose, no sodium pyruvate | Thermo Fisher Scientific | 11966–025 | |
| Other | Dialyzed fetal bovine serum (dFBS) | Thermo Fisher Scientific | 26400–044 | |
| Other | Dialyzed horse serum | Valley Biomedical | AS3053-DI | |
| Other | Horse serum, New Zealand origin | Thermo Fisher Scientific | 16050–130 | |
| Chemical compound, drug | *D*-(+)-Glucose solution | Sigma | G8769 | Working concentration 10 mM |
| Chemical compound, drug | Doxycycline hyclate | Sigma | D9891 | 300 ng/ml |
| Chemical compound, drug | Piericidin A | Santa Cruz | Sc-202287 | 0.5 µM |
| Chemical compound, drug | Antimycin A | Sigma | A8674 | 0.5 µM |
| Chemical compound, drug | Oligomycin A | Sigma | 75351 | 1 µM |
| Chemical compound, drug | BAM15 | TimTec | ST056388 | 0.5 µM (or as indicated) |
| Chemical compound, drug | Chloramphenicol | Sigma | C0378 | 20 µg/ml |
| Chemical compound, drug | Tunicamycin | Sigma | T7765 | 1 µg/ml |
| Chemical compound, drug | *L*-Aspartic acid | Sigma | A9256 | 10 mM |
| Chemical compound, drug | *L*-Asparagine monohydrate | Sigma | A8381 | 0.5 mM |
| Chemical compound, drug | Sodium pyruvate solution | Thermo Fisher Scientific | 11360–070 | 1 mM |
| Chemical compound, drug | Uridine | Sigma | U3003 | 200 µM |
| Chemical compound, drug | GCN2iB | Acme Bioscience (PMID:30061420) | | 0.5 µM |
| Chemical compound, drug | GSK2656157 | Cayman Chemical | 17372 | 0.25 µM |
| Chemical compound, drug | Sodium *DL*-Lactate-3,3,3-d3 | CDN Isotopes | D6556 | |
| Chemical compound, drug | Sodium pyruvate-$^{13}C_3$ | Sigma | 490717 | |
| Chemical compound, drug | Adenosine-$^{15}N_5$ 5'-Monophosphate | Sigma | 900382 | |
| Chemical compound, drug | Adenosine-$^{13}C_5$ 5'-Diphosphate | Toronto Research Chemicals | A281697 | |
| Chemical compound, drug | Adenosine-$^{13}C_{10}$ 5'-Triphosphate | Sigma | 710695 | |
| Chemical compound, drug | *L*-Aspartic acid-1,4-$^{13}C_2$ | Cambridge Isotope Laboratories | CLM-4455 | |
| Chemical compound, drug | Formic acid, LC/MS grade | Thermo Fisher Scientific | A117-50 | |

*Continued on next page*

*Continued*

| Reagent type (species) or resource | Designation | Source or reference | Identifiers | Additional information |
|---|---|---|---|---|
| Chemical compound, drug | Ammonium hydrogen carbonate, LC/MS grade | Sigma | 5.33005 | |
| Commercial assay or kit | TaqMan gene expression master mix | Thermo Fisher Scientific | 4369514 | |
| Commercial assay or kit | *Ubr3* mouse TaqMan probe | Thermo Fisher Scientific | Mm01328979_m1 | |
| Commercial assay or kit | *Atf3* mouse TaqMan probe | Thermo Fisher Scientific | Mm00476033_m1 | |
| Commercial assay or kit | *Ddit3* mouse TaqMan probe | Thermo Fisher Scientific | Mm01135937_g1 | |
| Commercial assay or kit | *Atf4* mouse TaqMan probe | Thermo Fisher Scientific | Mm00515325_g1 | |
| Commercial assay or kit | *Gdf15* mouse TaqMan probe | Thermo Fisher Scientific | Mm00442228_m1 | |
| Commercial assay or kit | *Ppp1r15a* mouse TaqMan probe | Thermo Fisher Scientific | Mm01205601_g1 | |
| Commercial assay or kit | *Trp53inp1* mouse TaqMan probe | Thermo Fisher Scientific | Mm00458142_g1 | |
| Commercial assay or kit | *TBP* human TaqMan probe | Thermo Fisher Scientific | Hs00427620_m1 | |
| Commercial assay or kit | *GDF15* human TaqMan probe | Thermo Fisher Scientific | Hs00171132_m1 | |
| Antibody | GCN2 (rabbit, polyclonal) | Cell Signaling Technology | 3302 | 1:500 dilution |
| Antibody | Phospho-GCN2 (Thr899) (rabbit, monoclonal) | Abcam | ab75836 | 1:500 dilution |
| Antibody | ATF4 (rabbit, monoclonal) | Cell Signaling Technology | 11815 | 1:500 dilution |
| Antibody | eIF2α (mouse, monoclonal) | Cell Signaling Technology | 2103 | 1:500 dilution |
| Antibody | Phospho-eIF2α (Ser51) (rabbit, monoclonal) | Cell Signaling Technology | 3597 | 1:500 dilution |
| Antibody | Actin (mouse, monoclonal) | Sigma | A4700 | 1:3000 dilution |

## Cell culture

C2C12 cells were obtained from ATCC (CRL-1772) and cell identity was authenticated by successful differentiation into myotubes and by RNA sequencing. The cells were used as-is for differentiation and were also infected with lentiviruses to generate stable cell lines expressing luciferase (control), *Lb*NOX, mito*Lb*NOX or NDI1 under a doxycycline-inducible promoter (TRE3G; Clontech, CA), as previously described (*Titov et al., 2016*). *Ndufa9*-knockout C2C12 cells were previously generated using CRISPR/Cas9 (*Vafai et al., 2016*). Primary mouse embryonic fibroblasts were obtained from Lonza (Morristown, NJ; M-FB-481) and primary human skeletal myoblasts from Thermo Fisher Scientific (San Jose, CA; A11440). Cells were kept in a 37C, 5% $CO_2$ incubator, except where otherwise indicated. Periodic testing for *Mycoplasma* was performed and was negative.

All cells were cultured in Dulbecco's Modified Eagle's Medium (DMEM). The base media formulation (Thermo Fisher Scientific; 11966–025) lacked glucose and sodium pyruvate and contained 4 mM *L*-Glutamine. Glucose (Sigma, St. Louis, MO; G8769) was always supplemented to a concentration of 10 mM.

Doxycycline-inducible myoblasts were cultured in media supplemented to 10% dialyzed fetal bovine serum (dFBS) (Thermo Fisher Scientific; 26400–044). Up until cells were seeded for experiments, the media also contained 1 μg/ml puromycin (Thermo Fisher Scientific; A11138-03) and 0.5 mg/ml geneticin (Thermo Fisher Scientific; 10131–035) to maintain selection of expression vectors.

*Ndufa9*-knockout C2C12 myoblasts, primary mouse embryonic fibroblasts, and primary human myoblasts not intended for differentiation were also cultured in media supplemented to 10% dFBS. For the *Ndufa9*-knockout cells, the standard culture media contained 1 mM sodium pyruvate (Thermo Fisher Scientific; 11360–070).

C2C12 myoblasts intended for differentiation were grown in media supplemented to 20% dFBS and kept sparse until seeded for experiments. When seeded cells had grown to confluence, they were switched to media supplemented to 2% dialyzed horse serum (Valley Biomedical, Winchester, VA; AS3053-DI) to induce differentiation. Human myoblasts intended for differentiation were seeded immediately after thaw directly at a confluent density in media supplemented to 2% horse serum, New Zealand origin (Thermo Fisher Scientific; 16050–130).

## Cell treatments for RNA isolation

Doxycycline-inducible C2C12 myoblasts were seeded for most treatments at 20,000 cells per well in 24-well plates with 0.5 ml/well of media without selection antibiotics. Only 2,000 cells per well were seeded for chloramphenicol treatment. Approximately 3 hr after initial seeding, an additional 0.5 ml/well of media supplemented with doxycycline (Sigma; D9891; dissolved in water) was dispensed to induce protein expression, such that the final concentration was 300 ng/ml.

24 hr after doxycycline addition, inhibitor treatments were started with complete replacement of the media to a final volume of 1 ml/well (still containing 300 ng/ml doxycycline). Piericidin A (Santa Cruz Biotechnology, Santa Cruz, CA; sc-202287) was used at 0.5 µM, Antimycin A (Sigma; A8674) at 0.5 µM, Oligomycin A (Sigma; 75351) at 1 µM, Tunicamycin (Sigma; T7765) at 1 µg/ml, Chloramphenicol (Sigma; C0378) at 20 µg/ml. All small-molecule inhibitors were dissolved in dimethyl sulfoxide (DMSO) (Sigma; D2650). Final DMSO concentrations did not exceed 1:1000.

When nutrients were supplemented during inhibitor treatments, they were introduced with the media at the start of treatment. *L*-Aspartic acid (Sigma; A9256) was supplemented to a concentration of 10 mM. Media supplemented with aspartate was titrated back to pH 7.4 with NaOH. *L*-Asparagine monohydrate (Sigma; A8381) was supplemented to a concentration of 0.5 mM. Uridine (Sigma; U3003) was supplemented to a concentration of 200 µM.

When eIF2α kinase inhibitors were used, they were introduced with the media at the start of treatment. The GCN2 inhibitor GCN2iB was custom synthesized (Acme Bioscience, Palo Alto, CA) based on the published recipe (*Nakamura et al., 2018*) and used at 0.5 µM. The PERK inhibitor GSK2656157 (Cayman Chemical, Ann Arbor, MI; 17372) was used at 0.25 µM. eIF2α kinase inhibitors were also dissolved in DMSO.

Primary mouse embryonic fibroblasts were seeded for treatments at 15,000 cells per well in 24-well plates with 1 ml/well of media, and primary human myoblasts (not intended for differentiation) were seeded at 30,000 cells per well in 12-well plates with 2 ml/well of media. Treatments were started 24 hr later with complete replacement of the media. Inhibitor and aspartate concentrations were as described above. Sodium pyruvate was supplemented to a concentration of 1 mM.

*Ndufa9*-knockout myoblasts were seeded at 20,000 cells per well in 24-well plates with 1 ml/well of media. 24 hr later, reductive stress was induced by replacing the culture media with media lacking sodium pyruvate. The aspartate concentration was as described above.

C2C12 myoblasts intended for differentiation were seeded at 50,000 cells per well in 24-well plates with 1 ml/well of media. Cells became fully confluent ~48 hr after seeding and were then switched to low-serum media. The media was fully replenished 48 hr after the switch to low-serum. If protein expression was required, the cells were transduced at this time with adenovirus expressing eGFP alone (Vector Biolabs, Malvern, PA; 1060-HT; 0.05 µl/well of ~5E12 VP/ml), or eGFP in combination with *Lb*NOX (0.125 µl/well of ~2E12 VP/ml) or with mito*Lb*NOX (0.05 µl/well of ~2E12 VP/ml). Adenovirus production was previously described (*Titov et al., 2016*). The media was fully replenished every 24 hr thereafter. After a total of 4 days in low-serum, differentiated myotubes were treated with mitochondrial inhibitors as described above. BAM15 (TimTec, Tampa, FL; ST056388) was used at 0.5 µM, except where otherwise indicated. For hypoxic pre-conditioning, myotubes were placed in a 37C, 5% $O_2$, 5% $CO_2$ incubator the night prior to the start of treatment and during treatment.

Human myoblasts intended for differentiation were seeded directly at a confluent density of 250,000 cells per well in 24-well plates with 1 ml/well of low-serum media. The media was fully

replenished 48 hr later and every 24 hr thereafter. After a total of 4 days, inhibitor treatments were started with complete replacement of the media. Inhibitor concentrations were as above.

Treatments intended for RNA isolation lasted 10 hr, except where otherwise indicated. At the end of the treatment, cells were lysed in 150 µl/well buffer RLT from the RNeasy kit (Qiagen, Germantown, MD; 74106). The lysates were immediately frozen at −80C until RNA isolation. RNA isolation was performed using the RNeasy kit or RNeasy96 kit (Qiagen; 74181) following the manufacturer's protocol.

## RNA sequencing and data analysis

The integrity of RNA intended for sequencing was assayed using Agilent Bioanalyzer 2100 or Advanced Analytical Fragment Analyzer. All tested RNA samples yielded optimal (10) or near optimal (>9) RIN/RQN values. 50 ng RNA per sample were submitted for preparation of sequencing libraries at the Broad Technology Labs based on the Smart-seq2 method (*Picelli et al., 2014*). The protocol included selection for polyadenylated RNA and was not strand-specific. Libraries were sequenced on two flow-cells of an Illumina NextSeq 500 instrument, generating $2 \times 37$ bp paired-end reads. Demultiplexed sequencing data has been deposited in the NCBI Gene Expression Omnibus (GEO) database under accession GSE132234.

Sequencing reads were pseudo-aligned using kallisto (*Bray et al., 2016*) (v. 0.44, with bias correction) to an index consisting of the mouse transcriptome (ENSEMBL v. 93; GRCm38.p6) and the sequences of the transgenes luciferase, *Lb*NOX, NDI1 and eGFP (previously reported *Titov et al., 2016*). Estimated counts were aggregated across the two flow-cells and summarized to the gene level in R using the tximport package (*Soneson et al., 2015*). Raw (un-normalized) gene level counts for all samples are available through the GEO record. Counts were then processed further in R using the DESeq2 package (*Love et al., 2014*). Gene fold-changes derived from RNA-seq were calculated after normalization for sequencing depth, as implemented in DESeq2.

Within a given subset of experimental conditions analyzed together, a gene was retained if there was at least one condition for which each replicate sample had at least 16 pre-normalization counts associated with the gene. Principal components analysis (PCA) was performed on the gene count data after regularized log2 transformation, as implemented in DESeq2, and mean centering. For the PCA visualization of all doxycycline-inducible myoblasts (*Figure 3F*) the data was also scaled. The jackstraw method (*Chung and Storey, 2015*) (10,000 resampling iterations, 10% of variables permuted in each iteration) was used to assign statistical significance for the association of each gene with each of the top principal components. Differential expression analyses based on GLM regression were performed in DESeq2. The design formulas and results are reported in *Supplementary file 1*. P-values are derived from the Wald test and were adjusted within each analysis with the method of Benjamini and Hochberg. Log2 fold-changes were shrunk using the apeglm shrinkage estimator (*Zhu et al., 2019*), as implemented in DESeq2.

*Cis*-regulatory analysis was performed with iRegulon (*Janky et al., 2014*) (v. 1.3) in Cytoscape (v. 3.7). For control myoblasts (*Figure 2B*), the input was the 500 genes with the most positive PC1 weights among genes with a PC1 jackstraw p-value<0.01, or, separately, the 500 genes with the most negative weights. For all doxycycline-inducible myoblasts (*Figure 3F*), the input was the 500 genes with the most positive PC2 weights among genes with a PC2 jackstraw p-value<0.0001. For enrichment of transcription factor binding motifs, the mouse gene list was used as-is. For enrichment of ChIP-seq binding peaks, the genes were first converted to the equivalent human genes, as this feature is not available directly for mouse. In all cases, the regulatory regions interrogated comprised 500 bp upstream of the transcription start site, as annotated in iRegulon. The seven species option was used in the rankings of transcription factor targets. The ROC threshold for AUC calculation was set at 1%.

Gene set enrichment analysis (GSEA) (*Subramanian et al., 2005*) was performed for REACTOME (*Croft et al., 2014*) pathways with a minimal size of 15 genes and a maximal size of 1000 genes. The input consisted of all genes with a PC1 jackstraw p-value<0.01 and pre-ranked by their PC1 weights. The gene list was converted to the equivalent human genes since REACTOME pathways are defined for human genes. GSEA was run using the fgseaMultilevel function in the R package fgsea (*Sergushichev, 2016*). The output was subset to pathways with a Benjamini-Hochberg corrected p-value<0.05. The overlap in leading edge genes between each pair of pathways was calculated and the overlap matrix was hierarchically clustered based on Euclidean distance with complete linkage.

The clustering guided manual selection of minimally redundant, representative gene sets to include in the visualization (*Figure 2C*).

## qPCR

Isolated RNA was annealed to random primers (Thermo Fisher Scientific; 48190–011) for 5 min at 70C, then reverse transcribed for 1 hr at 37C using M-MLV reverse transcriptase (Promega, Madison, WI; M1705) in the presence of RNase inhibitor (Thermo Fisher Scientific; 10777–019). The resulting cDNA was subjected to qPCR using TaqMan gene expression master mix (Thermo Fisher Scientific; 4369514) and TaqMan gene expression probes (see Key Resources Table for details) on a CFX96 instrument (Bio-Rad, Hercules, CA). Raw amplification cycle data was produced by the accompanying analysis software using default parameters. Cycle differences between tested conditions and the baseline condition were normalized against the reference gene *Ubr3* (for mouse samples) or *TBP* (for human samples), yielding $\Delta\Delta C_t$. Fold-changes were calculated as $2^{-\Delta\Delta Ct}$, but statistical testing was performed on the underlying $\Delta\Delta C_t$ values.

## [Lactate] and [pyruvate] in spent media

Cells were cultured and treated in the same way as for RNA isolation, except the final treatment media volume for myoblasts was 0.5 ml/well. 100 µl of the media in each well was collected 2 hr after the start of treatment and immediately frozen at −80C.

30 µl of the spent media was extracted by adding 117 µl acetonitrile and 20 µl water containing 1 mM Sodium *DL*-Lactate-3,3,3-d3 (CDN Isotopes, Pointe-Claire, Quebec, Canada; D6556) and 50 µM Sodium pyruvate-$^{13}C_3$ (Sigma; 490717). The samples were vortexed and left on ice for 10 min before being centrifuged at 21.1 k x g for 10 min. Finally, 100 µl of the supernatant was transferred to a glass vial and 10 µl was subjected to LC/MS analysis.

Media samples were separated using an Xbridge amide column (2.1 × 100 mm, 2.5 µm particle size) (Waters, Milford, MA), with mobile phase A: 5% acetonitrile, 20 mM ammonium acetate, 0.25% ammonium hydroxide, pH 9.0 and mobile phase B: 100% acetonitrile. The separation gradient was as follows: 85% B for 0.5 min, ramp to 35% B for 8.5 min, ramp to 2% B for 1 min, hold for 1 min, ramp to 85% B for 1.5 min, hold for 4.5 min. The flow rate was 220 µl/min for the first 14.6 min, then increased to 420 µl/min for the last 3.4 min. Mass spectrometry analysis was performed on a QExactive Plus instrument (Thermo Fisher Scientific) with polarity switching mode at 70,000 resolving power (at 200 m/z), a scan range of 70–1,000 m/z and an AGC target of 3E6.

Absolute concentrations for lactate and pyruvate were obtained based on a standard curve and used to report media [lactate]/[pyruvate] ratios and secreted [lactate]. For myoblasts, secreted [lactate] values were adjusted within each (transgene X treatment) combination based on the median [pyruvate] value, to account for variation in cell number.

## Cell extract metabolite profiling

Doxycycline-inducible myoblasts were seeded at 200,000 cells per dish in 6 cm dishes with 4 ml/dish of media without selection antibiotics. Approximately 3 hr later, 2 ml/dish were removed and replaced with 2 ml/dish of media containing doxycycline (final concentration 300 ng/ml). 24 hr after doxycycline addition, the media was fully replaced with 3 ml/dish fresh media (including doxycycline). 2 hr after the media replenishment, 1.5 ml/dish was removed and replaced with 1.5 ml/dish of media containing inhibitors at 2x the final concentrations, which were the same as described for RNA isolation.

Beginning at 1 hr after the start of treatments, dishes were individually removed from the incubator, placed on ice, the media was aspirated, cells were washed in 1 ml/dish ice-cold PBS and this was aspirated. Finally, 800 µl/dish metabolite extraction buffer was introduced, which consisted of 40% methanol:40% acetonitrile:20% water + 0.1M formic acid (Thermo Fisher Scientific; A117-50) to rapidly quench metabolism (*Lu et al., 2017*). The extraction buffer included 0.5 µM Adenosine-$^{15}N_5$ 5′-Monophosphate (Sigma; 900382), 1 µM Adenosine-$^{13}C_5$ 5′-Diphosphate (Toronto Research Chemicals, North York, Ontario, Canada; A281697), 5 µM Adenosine-$^{13}C_{10}$ 5′-Triphosphate (Sigma; 710695) and 2.5 µM *L*-Aspartic acid-1,4-$^{13}C_2$ (Cambridge Isotope Laboratories, Tewksbury, MA; CLM-4455). The cells were thoroughly scraped, the lysate was transferred to a fresh tube, vortexed briefly and placed on ice. 2 min later, 70 µl of 15% w/v ammonium bicarbonate (Sigma; 5.33005) was

pipetted into the tube to neutralize the pH, the tube was vortexed and placed at −20C until all extractions were completed. Then, all the tubes were spun at 4C at a speed of 21.1 k x g for 10 min to pellet cellular debris. 100 µl of the supernatant was transferred into a glass vial and 10 µl was immediately subjected to LC/MS analysis.

For myotubes, 200,000 cells were seeded per dish in 35 mm dishes with 3 ml/dish of media. The cells were differentiated for 4 days in low-serum media upon becoming confluent. Treatments were started as described for myoblasts. Metabolite extraction was performed as described for myoblasts except 500 µl/dish extraction buffer was used; the extraction buffer contained 1 µM labeled AMP, 5 µM labeled ADP, 25 µM labeled ATP and 10 µM labeled L-Aspartic acid; and 44 µl ammonium bicarbonate was used to neutralize the pH.

Extracted samples were separated using a ZIC pHILIC column (2.1 × 150 mm, 5 µm particle size) (EMD Millipore, Burlington, MA), with mobile phase A: 20 mM ammonium bicarbonate, and mobile phase B: 100% acetonitrile. The gradient began at 80% B, held for 0.5 min, ramped to 20% B for 20 min, held for 0.8 min, ramped to 80% B for 0.2 min, then held for 8.5 min. The flow rate was 150 µl/minute.

Progenesis QI software (Waters) was used to perform peak picking, peak alignment across samples, deconvolution of adduct peaks, peak intensity integration and intensity normalization across samples. An in-house metabolite retention time library of reference standards was used to identify the peaks based on accurate mass within a two ppm tolerance window and retention time within a 0.25 min tolerance window. Metabolite differential abundance analysis upon inhibitor treatment was performed on log2-transformed intensities using the R package limma. P-values are based on a moderated t-statistic.

The whole-cell NADH/NAD$^+$ ratio was calculated based on peak intensities. Absolute concentrations for AMP, ADP, ATP and aspartate were obtained based on a standard curve. The adenylate energy charge was calculated as: $\frac{[ATP]+\frac{1}{2}[ADP]}{[ATP]+[ADP]+[AMP]}$.

## Oxygen consumption rate (OCR) measurements

Doxycycline-inducible myoblasts were seeded at 15,000 cells per well in Seahorse 24-well cell culture microplates with 0.5 ml/well of media without selection antibiotics. 24 hr later, the media was replenished including doxycycline at a final concentration of 300 ng/ml. 24 hr after doxycycline addition, the media was replaced with 0.5 ml/well of the same media formulation except containing 25 mM HEPES-KOH instead of sodium bicarbonate. The cells were placed for 1 hr into a non-CO$_2$ controlled 37C incubator and then transferred to the XF24 Extracellular Flux Analyzer for OCR measurement. Each measurement was performed over 4 min after a 2 min mix and a 2 min wait. Inhibitors were introduced into the wells from the XF24 ports in 50 µl of media to the same final concentrations as described above for RNA isolation.

For myotube OCR measurement, myoblasts were seeded at 25,000 cells per well, became confluent within 24 hr and differentiated in low-serum media for 4 days. The rest of the protocol was as described above except the media volume in each well during the assay was 1 ml.

## Western blotting

Doxycycline-inducible LbNOX cells were seeded at 200,000 cells per dish in 6 cm dishes with 4 ml/dish of media without selection antibiotics. Approximately 3 hr later, 2 ml/dish were removed and replaced with 2 ml/dish of fresh media supplemented with either water, to avoid LbNOX expression, or doxycycline at a final concentration of 300 ng/ml, to induce expression. Inhibitor treatments were started 24 hr after water or doxycycline addition with complete replacement of the media (still including 300 ng/ml doxycycline where needed).

After 6 hr of treatment, each dish was individually removed from the incubator and handled as follows: the media in the dish was aspirated, the dish was briefly washed with 2 ml of ice-cold PBS and this was aspirated, 100 µl ice-cold 1X Laemmli SDS-sample buffer (Boston BioProducts, Ashland, MA; BP-111R) supplemented with protease/phosphatase inhibitor cocktail (Cell Signaling Technology, Danvers, MA; 5872S) was introduced, the dish was thoroughly scraped and the lysate was transferred into a tube and placed on ice. Because sample buffer is incompatible with standard assays for protein concentration, each experiment also included two spare, untreated dishes that were set up identically but lysed in 100 µl RIPA buffer (Boston BioProducts; BP-115) supplemented with

protease/phosphatase inhibitor cocktail and nuclease (Thermo Fisher Scientific; 88701). When all samples were collected, the tubes were vortexed, sample buffer lysates were boiled for 5 min at 95C and then all samples were frozen at −80C.

When lysates were thawed, protein concentration was determined in the RIPA lysates using the DC protein assay kit (Bio-Rad; 5000112). Volumes of sample buffer lysates corresponding to 30 µg of protein were boiled for 5 min at 95C, resolved on 4–12% Tris-Glycine mini gels (Thermo Fisher Scientific; XP04120BOX) and transferred to nitrocellulose membranes (Bio-Rad; 1704159) using a 2 hr wet transfer. Membranes were blocked for 30 min with Intercept TBS blocking buffer (LI-COR Biosciences, Lincoln, NE; 927–60001).

To probe the complete ISR pathway on the same membrane, as in *Figure 4L*, the membrane was first probed overnight at 4C with an anti-GCN2 antibody (rabbit, polyclonal; Cell Signaling Technology; 3302) at a dilution of 1:500 in Intercept T20 TBS antibody diluent (LI-COR Biosciences; 927–65001). The following day, the membrane was washed three times with TBST for 5 min each and then incubated for 1 hr at room temperature (RT) with IRDye 800CW goat anti-rabbit IgG secondary antibody (LI-COR Biosciences; 926–32211) at a dilution of 1:10,000 in the antibody diluent. The membrane was again washed 3 times for 5 min each and then scanned for infrared signal on the Odyssey imaging system (LI-COR Biosciences). The membrane was then incubated for 15 min at RT with Restore western blot stripping buffer (Thermo Fisher Scientific; 21059), blocked for 30 min and probed overnight at 4C with an anti-Phospho-GCN2 (Thr899) antibody (rabbit, monoclonal; Abcam; ab75836) at a dilution of 1:500. The following day, the membrane was washed, incubated with anti-rabbit secondary antibody and imaged. After this, the membrane was probed for 2 hr at RT with an anti-ATF4 antibody (rabbit, monoclonal; Cell Signaling Technology; 11815) at a dilution of 1:500 along with an anti-Actin antibody (mouse, monoclonal; Sigma; A4700) at a dilution of 1:3000. The membrane was then washed, incubated with both the anti-rabbit secondary antibody and an IRDye 680RD goat anti-mouse IgG secondary antibody (LI-COR Biosciences; 926–68070) at a dilution of 1:10,000 and imaged. The membrane was then probed overnight at 4C with an anti-phospho-eIF2$\alpha$ (Ser51) antibody (rabbit, monoclonal; Cell Signaling Technology; 3597) at a dilution of 1:500. The next day, the membrane was washed, incubated with anti-rabbit secondary antibody and imaged. Finally, the membrane was probed for 2 hr at RT with an anti-eIF2$\alpha$ antibody (mouse, monoclonal; Cell Signaling Technology; 2103) at a dilution of 1:500, washed, incubated with anti-mouse secondary antibody and imaged. Thus, the exact same bands were probed for both total- and phospho-eIF2$\alpha$, and the corresponding signals obtained at separate wavelengths. Note, when GCN2 was not examined, as in *Figure 2B*, the procedure began directly with probing for ATF4 and Actin. Band intensities were quantified using Image Studio Lite (LI-COR Biosciences).

## Cell proliferation

Doxycycline-inducible myoblasts were seeded at 5,000 cells per well in 24-well plates with 0.5 ml/well of media without selection antibiotics. Approximately 3 hr later, an additional 0.5 ml/well of media supplemented with doxycycline was dispensed in each well, such that the final concentration was 300 ng/ml. Inhibitor treatments were started 24 hr later with complete replacement of the media (still including 300 ng/ml doxycycline). Baseline counts for each condition were collected immediately after the start of treatment from separate wells. Final counts were collected 24 hr after the start of treatment. Counts were obtained using a Z2 Coulter Particle Count and Size Analyzer (Beckman Coulter, Brea, CA). The proliferative rate was calculated as the number of cell doublings between final and baseline counts.

## Acknowledgements

We thank the Klarman Cell Observatory at the Broad Institute for support and technical assistance in the early stages of the project, and members of the Mootha laboratory for discussions, advice and comments on the manuscript. VKM is an Investigator of the Howard Hughes Medical Institute.

## Additional information

### Competing interests

Eran Mick: listed as an inventor on a patent filed by Massachusetts General Hospital on the therapeutic uses of GCN2 inhibitors. Owen S Skinner: a paid consultant for Proteinaceous Inc. Vamsi K Mootha: listed as an inventor on a patent filed by Massachusetts General Hospital on the therapeutic uses of GCN2 inhibitors, and a paid advisor to Janssen Pharmaceuticals and 5am Ventures. The other authors declare that no competing interests exist.

### Funding

| Funder | Grant reference number | Author |
|---|---|---|
| National Institutes of Health | R35GM122455 | Vamsi K Mootha |
| Marriott Foundation | | Vamsi K Mootha |
| Ruane Family Foundation | | Vamsi K Mootha |
| Howard Hughes Medical Institute | | Vamsi K Mootha |
| Howard Hughes Medical Institute | Graduate Student Fellowship | Eran Mick |
| National Institutes of Health | 1F32GM133047-01 | Owen S Skinner |
| Swiss National Science Foundation | Postdoctoral Fellowship | Alexis A Jourdain |

The funders had no role in study design, data collection and interpretation, or the decision to submit the work for publication.

### Author contributions

Eran Mick, Conceptualization, Data curation, Formal analysis, Validation, Investigation, Visualization, Methodology, Writing - original draft, Project administration, Writing - review and editing; Denis V Titov, Conceptualization, Resources, Methodology; Owen S Skinner, Rohit Sharma, Alexis A Jourdain, Resources, Data curation, Methodology; Vamsi K Mootha, Conceptualization, Supervision, Funding acquisition, Writing - original draft, Project administration, Writing - review and editing

### Author ORCIDs

Eran Mick (iD) https://orcid.org/0000-0002-7299-808X
Denis V Titov (iD) https://orcid.org/0000-0001-5677-0651
Owen S Skinner (iD) https://orcid.org/0000-0002-5023-0029
Alexis A Jourdain (iD) https://orcid.org/0000-0001-5321-6938
Vamsi K Mootha (iD) https://orcid.org/0000-0001-9924-642X

### Decision letter and Author response

Decision letter https://doi.org/10.7554/eLife.49178.sa1
Author response https://doi.org/10.7554/eLife.49178.sa2

## Additional files

### Supplementary files

• Supplementary file 1. Analysis of RNA sequencing data (counts, PCA, differential expression). Related to *Figure 2A, D-F*, *Figure 2—figure supplement 1A*, *Figure 3F*, *Figure 5A,I*, *Figure 5—figure supplement 1D*.

• Supplementary file 2. iRegulon *cis*-regulatory analysis. Related to *Figure 2B* and *Figure 3F*.

• Transparent reporting form

## Data availability

Sequencing data have been deposited in GEO under accession code GSE132234.

The following dataset was generated:

| Author(s) | Year | Dataset title | Dataset URL | Database and Identifier |
|---|---|---|---|---|
| Mick E, Titov DV, Skinner OS, Sharma R, Jourdain AA, Mootha VK | 2020 | Distinct mitochondrial defects trigger the integrated stress response depending on the metabolic state of the cell | https://www.ncbi.nlm.nih.gov/geo/query/acc.cgi?acc=GSE132234 | NCBI Gene Expression Omnibus, GSE132234 |

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
