## [Decision Letter]

**Acceptance summary:**

We find it of importance that your study succeeded in uncovering the proximal signals sensed by the Integrated Stress Response and the signalling pathways involved during a reduction in cellular respiration. We feel that this paper is an important addition to the literature of electron transport chain defects and their cellular consequences.

**Decision letter after peer review:**

Thank you for submitting your article "Distinct mitochondrial triggers of the integrated stress response in proliferating and post-mitotic cells" for consideration by *eLife*. Your article has been reviewed by three peer reviewers, one of whom is a member of our Board of Reviewing Editors, and the evaluation has been overseen by David Ron as the Senior Editor. The following individuals involved in review of your submission have agreed to reveal their identity: Peter Walter (Reviewer #2). I am happy to tell you that all three reviewers were excited by your work and would like to see it published in *eLife* following some requested revisions.

The reviewers have discussed the reviews with one another and the Reviewing Editor has drafted this decision to help you prepare a revised submission.

Summary:

The manuscript by Mick et al. addresses a highly important and interesting question – what exactly is sensed by the integrated stress response (ISR) when mitochondria are dysfunctional. They interrogate the proximal signals sensed by the ISR in muscle cells (differentiated and undifferentiated) in culture treated with various chemical inhibitors of the respiration process at different steps and with different metabolic consequences. The main finding of the paper is that there are at least two different ways for mitochondrial dysfunction to activate the ISR. The most mechanistic insight is gained in proliferating myoblasts where they show that the ISR is triggered by impaired NADH oxidation largely due to an aspartate deficiency and activation of the eIF2α kinase GCN2. Surprisingly, myotubes only triggered the response upon ATP synthase inhibition although it was not the drop in ATP that was sensed. It is still not quite clear what exactly is being sensed in this pathway. In summary, this paper brings a first report of the mechanism of sensing of the ISR of mitochondrial stress and as such is an important addition to the literature of ETC defects and their cellular consequences.

Essential revisions:

1) A central issue with the paper is that while the authors beautifully show that upon complex I inhibition aspartate depletion occurs and while they also nicely show that GCN2 is required for the induction of the ISR upon complex I inhibition – the demonstrating that the link is causal is still missing. Hence from their work it is not clear that GCN2 senses the aspartate levels (although this is very plausible) or whether this is merely a correlation. We encourage the authors to experiment to show causality or else reword the manuscript. One potential experiment to support a direct sensing by GCN2 is to show how piericidin and aspartate addition affect activation of GCN2 directly (by phosphorylation level).

2) The authors show the induction of an *Atf4*-dependent transcriptional response reminiscent of the ISR and further confirm that inhibition of the kinase GCN2 (but not PERK) can block the piericidin-induced induction of downstream *Atf4* targets, such as *Atf3* and *Ddit3*. However, more evidence is needed that this is a true canonical ISR response, to differentiate it from ISR-independent transcriptional induction of *Atf4*. While the authors show that indeed, eIF2α is phosphorylated and ATF4 protein is induced upon piericidin treatment, this evidence is lacking for the other inhibitors or conditions tested. Figure 2E, 3E, 4F, I, J for example, only use the transcriptional induction of one or two targets to claim ISR modulation. Moreover, the authors overlooked a major result of ISR activation, namely global downregulation of protein synthesis. This can easily be addressed with puromycin or 35S-incorporation Western blot assays.

3) In Figure 4A the reduction in phosphorylated eIF2α after *Lb*NOX expression is not clear. In addition it seems that *Lb*NOX itself triggers some basal eIF2α activation. If the authors are certain that there is a reduction in phosphorylation (not relative but absolute) that they should quantify this and show quantification or modify their description of the data.

4) Statistical analysis are missing for almost all the graphs. Please make sure to add this.

5) A suggestion for authors (not necessary for acceptance) is that an analysis of protein synthesis rates in the two cell types could nicely illustrate the relative sensitivity of myoblasts versus myotubes to aspartate deficiency (Discussion paragraph six). A puromycin blot, 35S incorporation assay or similar should suffice.

---

## [Author Response]

Essential revisions:1) A central issue with the paper is that while the authors beautifully show that upon complex I inhibition aspartate depletion occurs and while they also nicely show that GCN2 is required for the induction of the ISR upon complex I inhibition – the demonstrating that the link is causal is still missing. Hence from their work it is not clear that GCN2 senses the aspartate levels (although this is very plausible) or whether this is merely a correlation. We encourage the authors to experiment to show causality or else reword the manuscript. One potential experiment to support a direct sensing by GCN2 is to show how piericidin and aspartate addition affect activation of GCN2 directly (by phosphorylation level).

We thank the reviewers for this thoughtful suggestion. We investigated the nature of GCN2 sensing of the amino acid deficiency upon complex I inhibition in myoblasts by examining GCN2 autophosphorylation at Threonine-898 as a readout of kinase activation. The results show that GCN2 is activated by complex I inhibition and that the proximal signal is likely the depletion of asparagine that results from the aspartate deficiency.

We base our conclusion on the observation that aspartate addition partially abrogates GCN2 autophosphorylation (Figure 4L), in line with a partial rescue of cellular asparagine levels (Figure 4E). Asparagine alone, on the other hand, abrogates GCN2 autophosphorylation with no additive effect of aspartate (Figure 4L). Importantly, most mammalian cells cannot convert asparagine back to aspartate.

That GCN2 did not directly sense the profound aspartate depletion seems surprising. This result may be explained by the fact that while the K_M_ values for aspartate and asparagine of their respective cytosolic tRNA synthetases are similar (~15-30μM), the cellular concentrations of these amino acids are orders of magnitude apart. Aspartate is estimated at ~5-10mM whereas asparagine is much closer to the K_M_, at ~100-200μM. Thus, even a large fold-change drop in aspartate would not compromise its cognate tRNA charging but a smaller drop in asparagine would, activating GCN2. The K_M_ for aspartate of asparagine synthetase is much higher (~1mM), so aspartate deficiency readily depletes asparagine. Nevertheless, it remains possible the aspartate deficiency can impact ISR-related gene expression independently of asparagine at a later time. We now include these ideas in the Discussion section.

2) The authors show the induction of an Atf4-dependent transcriptional response reminiscent of the ISR and further confirm that inhibition of the kinase GCN2 (but not PERK) can block the piericidin-induced induction of downstream Atf4 targets, such as Atf3 and Ddit3. However, more evidence is needed that this is a true canonical ISR response, to differentiate it from ISR-independent transcriptional induction of Atf4. While the authors show that indeed, eIF2α is phosphorylated and ATF4 protein is induced upon piericidin treatment, this evidence is lacking for the other inhibitors or conditions tested.

We now show the mitochondrial inhibitors we used (piericidin, antimycin, oligomycin) all induce eIF2α phosphorylation and ATF4 protein accumulation (Figure 2—figure supplement 1B). Thus, canonical ISR signaling is engaged in each of these cases, as has typically been observed in studies on the ISR in mitochondrial stress (e.g., Kim et al., 2013 (PMID: 23202295); Bao et al., 2016; Wang et al., 2016 (PMID: 27708226); Guo et al., 2020). However, this does not rule out potentially independent transcriptional effects that impact individual transcripts, including *Atf4*. The results we report indeed mostly refer to the overall ISR-related gene expression program observed upon mitochondrial inhibition, which is likely shaped by multiple layers of regulation. We now make this point more explicit in describing our results.

Figure 2E, 3E, 4F, I, J for example, only use the transcriptional induction of one or two targets to claim ISR modulation.

– Figure 2E illustrates the gradation of ISR gene expression across the inhibitors using selected ATF4/DDIT3 targets from the RNA-seq data. However, this follows panels A-D that establish this pattern using PCA, gene set enrichment and transcription factor binding site analyses applied to the full RNA-seq dataset. Moreover, Figure 2—figure supplement 1A shows many additional targets that behave similarly to those shown in Figure 2E (and some that do not, as we point out). Finally, all ATF4/DDIT3 targets are annotated in Supplementary file 1 and can be examined individually. New Figure 2—figure supplement 1B now shows p-eIF2α and ATF4 protein levels as well.

– Similarly, Figure 3E shows one ISR target for illustration purposes but the PCA plot in Figure 3F considers the full RNA-seq dataset and demonstrates global attenuation of ISR gene expression by *Lb*NOX expression. The effects on each transcript can be examined in Supplementary file 1. We have shown the *Lb*NOX rescue is reflected at the level of p-eIF2α and ATF4 protein (previously Figure 4A, now Figure 4L and Figure 4—figure supplement 1L).

– Figure 4 and Figure 4—figure supplement 1 show that aspartate attenuates the induction of ISR targets upon piericidin treatment across four individual transcripts, and we have now extended this result to mouse and human primary cells as well (Figure 4J and Figure 4—figure supplement 1J). Figure 4L now shows the levels of p-GCN2, ATF4 and p-eIF2α across all the conditions that modulate ISR activation by piericidin (Asp, Asn, Asp+Asn, *Lb*NOX and GCN2iB). We demonstrate overall concordance between the effects on ATF4 protein and on transcriptional targets at the same time-point (Figure 4M).

Moreover, the authors overlooked a major result of ISR activation, namely global downregulation of protein synthesis. This can easily be addressed with puromycin or 35S-incorporation Western blot assays.

Acute mitochondrial inhibition has consistently been observed to downregulate global protein synthesis, including in C2C12 cells (e.g., Yeh et al., 1992 (PMID: 1643074); Zheng et al., 2016 (PMID: 27008180); Guo et al., 2020). We did not overlook this consequence of ISR activation and noted it in the Introduction. However, we reasoned that measurement of protein synthesis rates would not provide an interpretable readout of the ISR across our conditions of interest since additional factors fundamentally confound it. Specifically, we show ETC inhibition in myoblasts leads to severe amino acid deficiency, effects on energy metabolism and proliferative arrest, each of which is expected to impact protein synthesis independently of the ISR.

An instructive demonstration of these confounding effects is provided by two recent studies. Nakamura et al., 2018, applied GCN2 inhibitors to cells treated with asparaginase. GCN2 inhibition abrogated eIF2α phosphorylation and ATF4 accumulation – i.e., shut down ISR signaling – but it failed to restore protein synthesis, presumably because the underlying asparagine deficiency remained unresolved (Figure S4 in that paper). Guo et al., 2020 observed that HRI knockout attenuated ATF4 accumulation in oligomycin-treated cells but it nevertheless failed to restore protein synthesis (Ext. Figure 5 in that paper), which the authors ascribed to the effects on energy metabolism.

Similarly, in our study, it would be impossible to tell whether differences in protein synthesis rates among inhibitors or interventions arose due to effects on ISR signaling, amino acid levels, energy metabolism or overall control of proliferation and growth. For this reason, we believe that in the specific setting of mitochondrial dysfunction, protein and transcriptional markers of the ISR are the only informative, direct readouts of the response.

3) In Figure 4A the reduction in phosphorylated eIF2α after LbNOX expression is not clear. In addition it seems that LbNOX itself triggers some basal eIF2α activation. If the authors are certain that there is a reduction in phosphorylation (not relative but absolute) that they should quantify this and show quantification or modify their description of the data.

We agree the effects on p-eIF2α are less pronounced compared to GCN2, ATF4 or transcriptional ISR targets, both from a technical perspective (narrow dynamic range) and from a biological perspective (integrator of multiple, possibly compensatory, inputs). We have now emphasized these points in the Results and in the Discussion. In the case of *Lb*NOX, however, we do consistently observe attenuation of phospho/total- eIF2α following piericidin treatment, and no basal phosphorylation, based on quantification of 4 western blots (Figure 4—figure supplement 1L). These blots were performed with a more accurate protocol that measured the ratio from the same protein band using two-color fluorescent antibodies.

4) Statistical analysis are missing for almost all the graphs. Please make sure to add this.

We have added statistical analysis for key results.

5) A suggestion for authors (not necessary for acceptance) is that an analysis of protein synthesis rates in the two cell types could nicely illustrate the relative sensitivity of myoblasts versus myotubes to aspartate deficiency (Discussion paragraph six). A puromycin blot, 35S incorporation assay or similar should suffice.

Unfortunately, because aspartate is not an essential amino acid supplied in the culture media, we cannot directly control the magnitude of its deficiency. As we show in Figure 4D and Figure 5G, aspartate is depleted much more in myoblasts treated with piericidin (~16-fold) than in myotubes (~2-fold). Thus, a comparison of the resulting protein synthesis rates would not actually report on the relative sensitivity to aspartate deficiency of myoblasts versus myotubes. We note studies involving deprivation of essential amino acids, such as leucine, have shown that myoblasts downregulate protein synthesis faster and to a greater extent than myotubes in these circumstances (Talvas et al., 2006; Deval et al., 2008).